# Augmenting Research Ideation with Data: An Empirical Investigation in Social Science

## Abstract

Large Language Models (LLMs) show strong potential for generating novel research ideas, yet such ideas often struggle with feasibility and effectiveness. In this paper, we investigate whether augmenting LLMs with relevant data during the ideation process can improve idea quality. Our framework integrates data at two stages: (1) incorporating metadata during idea generation to guide models toward more feasible concepts, and (2) introducing an automated preliminary validation step during idea selection to assess the empirical plausibility of hypotheses within ideas. For the ease of data collection and expert evaluation, we conduct experiments in the social science domain, with a specific focus on climate negotiation topics. Expert evaluation shows that metadata improves the feasibility of generated ideas by 20%, while automated validation improves the overall quality of selected ideas by 7%. Beyond assessing the quality of LLM-generated ideas, we conduct a human study to examine whether these ideas, augmented with related data and preliminary validation, can inspire researchers in their own ideation. Participants report that the LLM-generated ideas and validation are highly useful, and the ideas they propose with such support are proven to be of higher quality than those proposed without assistance. Our findings highlight the potential of data-augmented research ideation and underscore the practical value of LLM-assisted ideation in real-world academic settings.

## 1 Introduction

Recent advances in Large Language Models (LLMs) have shown promise in generating research ideas, with some studies suggesting that these ideas can exhibit greater novelty than those proposed by human experts (Si et al., 2024; Yamada et al., 2025). However, many LLM-generated ideas suffer from practical limitations: they may be infeasible to implement, lack suitable datasets for validation, or have uncertain effectiveness. For instance, an LLM might propose investigating *"the impact of diplomats' childhood environmental experiences on their bargaining positions in UN climate negotiations"*, which is an interesting idea but lacks available data for empirical analysis.

A potential reason is that current ideation methods mainly rely on literature, without guidance from empirical data (Yang et al., 2024a; Li et al., 2024a). Intuitively, if LLMs are provided with relevant datasets, they could better equipped to generate empirically grounded research ideas: those that are not only novel but also feasible for experimentation. Just as human researchers navigate trade-offs between theoretical ambition and empirical tractability when developing research ideas, LLMs could benefit from this balancing act when data is available. For example, if the LLM is aware of the existence of records on climate conference attendance, it might propose a more feasible study, like *"how the professional backgrounds of diplomats influence their countries' emission reduction commitment ambitions."*

Besides guiding LLMs towards **feasible** ideas, data also enables preliminary validation of hypotheses within ideas. With access to relevant datasets, LLMs can write code to analyze the data and perform reasoning to assess whether the hypotheses are supported by the available evidence. While this validation is preliminary and does not guarantee sound conclusions, it provides valuable signals regarding whether the ideas are likely to be **effective**. Although recent works involve data in experiment execution of end-to-end scientific discovery (Jansen et al., 2025; Lu et al., 2024), to our

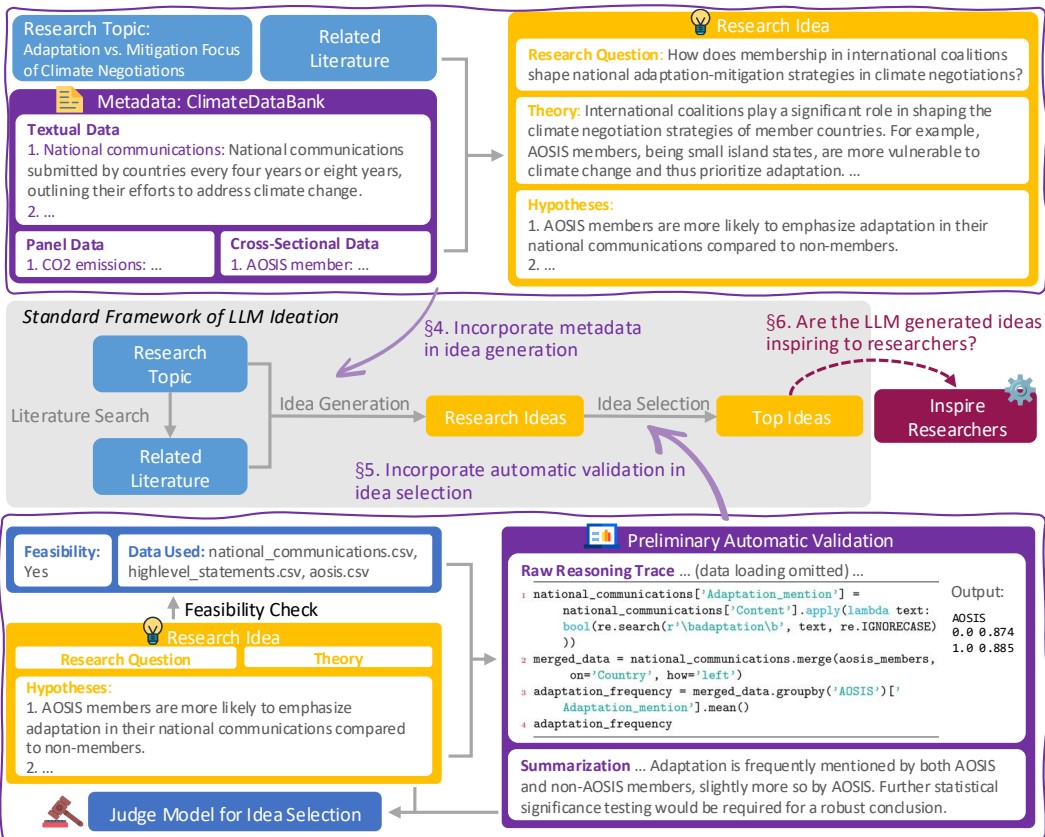

Figure 1: Overview of the data-augmented LLM ideation framework. Compared to the standard ideation framework (center), our approach integrates metadata into the idea generation stage (top) and adds preliminary automatic validation to the idea selection stage (bottom).

best knowledge, none investigate the role of data during ideation, nor quantify its influence through rigorous expert evaluation.

In this paper, we address this gap by investigating *whether augmenting LLMs with data during the ideation process can enhance the quality of generated ideas*. As shown in the center of Figure 1, the standard framework for LLM ideation consists of three stages: literature search, idea generation, and idea selection. We enhance this framework by incorporating data at two key stages: (1) During idea generation, we provide metadata, such as dataset descriptions, to guide models toward feasible research directions; and (2) during idea selection, we integrate automatic validation to account for the empirical plausibility of the proposed hypotheses within ideas.

We conduct experiments in the domain of social science. Because social science research often deals with complex, context-dependent phenomena, access to high-quality data is essential for grounding theories in observable evidence and distinguishing between ideas that are merely interesting and those that are empirically testable. For the ease of data collection and expert evaluation, we concentrate on topics related to climate negotiations, and gather relevant datasets into a unified CLIMATE-DATABANK. Expert evaluation reveals that incorporating metadata improves the feasibility by 20% and the expected effectiveness by 18%, and the overall quality of ideas selected with validation is rated 7% higher than those without validation.

Beyond assessing the quality of generated ideas, we explore *whether LLM-generated ideas, along with their related data and validation processes, can inspire human researchers to develop their own ideas*. In a study with 23 researchers, we find that compared to traditional idea creation aided only by the Internet, participants propose ideas of higher quality when given the reference of LLM-generated ideas. Feedback from participants indicates that LLM-generated ideas and validation processes are

very helpful, with some researchers using them as starting points for further refinement, which helps broaden their thinking.

Our contributions are as follows: (1) We novelly investigate the role of data in LLM ideation, and introduce a data-augmented ideation framework that integrates dataset metadata in idea generation and automatic validation in idea selection. (2) Both automatic and expert evaluation indicate that metadata and automatic validation improve the quality of generated ideas, particularly in feasibility and expected effectiveness. (3) Human study reveals that LLM-generated ideas can inspire researchers to propose higher-quality ideas, demonstrating the practical value of LLM ideation. (4) We construct the CLIMATEDATABANK to support future work in data-driven ideation.

## 2 RELATED WORK

Table 1: Comparison with existing works on research ideation and automated scientific discovery.

| Work | Primary Focus | Data Integration Stage | Help to Researchers |
|------|---------------|------------------------|---------------------|
| Si et al. (2024) | Generate research ideas | ✗ | Not explored |
| Baek et al. (2024) | Generate research ideas | ✗ | Not explored |
| Jansen et al. (2025) | End-to-end scientific discovery | Experiment execution | Not explored |
| Lu et al. (2024) | End-to-end scientific discovery | Experiment execution | Not explored |
| Ours | Generate more feasible and effective ideas | Idea generation and selection | Inspire researchers in their own ideation process |

**Research Idea Generation** There is growing interest in leveraging LLMs for research idea generation, either as a standalone task (Si et al., 2024; Baek et al., 2024) or as part of an end-to-end automated research pipeline (Li et al., 2024b; Lu et al., 2024; Jansen et al., 2025). The first category of work focuses on enhancing literature search and idea formulation, typically generating ideas grounded in prior work (Wang et al., 2024; Yang et al., 2024a). The second category of work proposes more comprehensive frameworks that encompass later research stages, such as experiment design, execution, and paper writing.

Table 1 summarizes how our work differs from prior studies. The focus of our work aligns closely with the first category, but novelly incorporates data into the ideation process, aiming to generate more feasible and effective ideas. While our work also involves code generation and execution for hypothesis validation, this is not intended as rigorous experiment execution, but serves as a preliminary signal to support idea selection, which differs from the second category. Moreover, our work investigates how the generated ideas can support human researchers, which is rarely covered in previous research.

**Hypothesis Generation** A related but distinct line of work focuses on hypothesis generation, where models generate hypotheses to explain phenomena given access to data, like inducing rules from observations (Zhong et al., 2023; Qiu et al., 2024). Studies in this area explore data-driven methods (Majumder et al., 2024a; Zhou et al., 2024) or integrate literature with data (Liu et al., 2024a), but their goal is to uncover patterns in existing datasets, contrasting with our objective of generating high-quality research ideas.

## 3 DATA COLLECTION

**CLIMATEDATABANK Construction** We first collect data related to climate negotiations, constructing the CLIMATEDATABANK to facilitate the following experiments. Our process begins with a comprehensive literature review, identifying important and commonly used datasets. We collect common variables from World Bank Open Data[1], and other datasets from their original sources.

The CLIMATEDATABANK is composed of three primary types of data: (1) Textual data, which includes documents such as national communications and high-level statements issued by various countries, enabling both qualitative analysis and text mining. (2) Panel data, such as the Gross Domestic Product (GDP) of each country over time, facilitating longitudinal analysis of trends over

---

[1]https://data.worldbank.org/

multiple years. (3) Cross-sectional data, capturing static attributes such as membership in the Alliance of Small Island States (AOSIS), with all values standardized to the year 2025 for consistency.

CLIMATEDATABANK contains 22 datasets in total, each stored in CSV format for ease of access. The full list of datasets and corresponding data descriptions is in Appendix Table 8.

**Reference Paper Collection** During the literature review, we also collect papers with clear hypotheses and replicable data. After manually reviewing 103 papers, we identify 8 papers that meet these criteria, as shown in Appendix Table 9. These papers, along with the corresponding data, are used in §5, where models are asked to validate the hypotheses in these papers, and rank the ground-truth ideas among LLM-generated ideas.

**Research Topics** We generate 10 climate negotiation-related research topics (shown in Appendix Table 10) using GPT-4o (Hurst et al., 2024), and manually verify them to ensure their quality.

Given the cost of constructing the data bank and recruiting domain experts to conduct human evaluation, we restrict experiments to research topics related to climate negotiations. However, the framework we propose can be broadly applied to diverse quantitative social science studies, where researchers need to build a theory for a phenomenon and test hypotheses to support the theory.

## 4 INCORPORATING METADATA IN IDEA GENERATION

This section explores the role of metadata in idea generation. We first describe how social science research ideas are structured and generated, then explain how metadata is integrated into the generation process. Finally, we present both automatic and human evaluation results.

### 4.1 SOCIAL SCIENCE IDEA GENERATION

A typical social science research idea consists of three components: a research question $r_q$, a theory $t_h$, and several hypotheses $\boldsymbol{h}$ (Powner, 2014; King et al., 1994). As illustrated in the top right example of Figure 1, the research question guides the study by identifying the central issue to be explored. The theory speculates on the answer to the research question and explains why the proposed answer is reasonable. The hypotheses identify observable implications of the theory, i.e., things we would observe if the theory is correct.

In the standard process, given a research topic $t$, LLMs first conduct a literature search and retrieve related literature $L$, and then generate research ideas with the components $(r_q, t_h, \boldsymbol{h})$ through the idea generation stage. The generated ideas are then passed to the idea selection stage to select the top-ranked ideas.

### 4.2 INCORPORATING METADATA INTO IDEA GENERATION

Figure 1 (top) shows how we incorporate metadata, which is concise dataset descriptions, into the idea generation stage along with the topic and related literature. Each metadata entry summarizes a dataset in CLIMATEDATABANK with one or two sentences, including information like the meaning of key variables, temporal coverage, and spatial scope. The prompt informs LLMs that *here are existing data related to this topic*, without strict restriction on using the provided data. This ensures that models can balance theoretical creativity with empirical feasibility by themselves.

By exposing models to metadata early, we encourage data-informed ideation where the feasibility of measurement is considered. Note that in this stage, we provide only the metadata, not the real content of the data, avoiding models from conducting data dredging by finding patterns in data and disguising them as hypotheses.

### 4.3 EXPERIMENTAL SETUP

**Methods** We experiment with three prevalent research idea generation methods: AI-Researcher (Si et al., 2024), GPT-Researcher (Elovic, 2023), and Chain-of-Ideas (Li et al., 2024a). Each method first retrieves relevant literature and then generates ideas, with detailed descriptions in Appendix C. We preserve each method's original design while adding metadata to the idea generation prompt.

Table 2: ELO scores of ideas generated with (w.) and without metadata. Better results are in bold.

| Method | w. Metadata | Significance | Novelty | Feasibility | Exp. Effectiveness | Average |
|---|---|---|---|---|---|---|
| | | Gemini-1.5-Pro as the Judge | | | | |
| AI-Researcher | ✗ | 902 | **933** | 1047 | 951 | 958 |
| | ✓ | **938** | 931 | **1098** | **997** | **991** |
| GPT-Researcher | ✗ | 1019 | 1000 | 1045 | 1015 | 1020 |
| | ✓ | **1073** | **1021** | **1183** | **1134** | **1103** |
| Chain-of-Ideas | ✗ | 974 | 1025 | **822** | 915 | 934 |
| | ✓ | **1094** | **1091** | 805 | **988** | **995** |
| | | Claude-3.5-Sonnet as the Judge | | | | |
| AI-Researcher | ✗ | **870** | **968** | 1060 | 881 | 945 |
| | ✓ | 855 | 859 | **1152** | **918** | **946** |
| GPT-Researcher | ✗ | 931 | **972** | 1076 | 928 | 977 |
| | ✓ | **1000** | 903 | **1228** | **1085** | **1054** |
| Chain-of-Ideas | ✗ | 1066 | 1118 | **768** | 1012 | 991 |
| | ✓ | **1278** | **1180** | 716 | **1176** | **1088** |

Table 3: Human evaluation of ideas generated by GPT-Researcher with and without metadata (%). Significance level is computed with McNemar' test (* for $p < 0.1$, ** for $p < 0.05$, and *** for $p < 0.01$).

| | w. Metadata | Tie | w/o Metadata | Significance Level |
|---|---|---|---|---|
| Significance | **38.8** | 22.4 | **38.8** | - |
| Novelty | 42.6 | 14.0 | **43.4** | - |
| Feasibility | **46.5** | 27.1 | 26.4 | *** |
| Exp. Effectiveness | **51.2** | 16.3 | 32.5 | ** |
| Overall | **43.4** | 14.7 | 41.9 | - |

For each research topic, we generate 50 ideas per method, and select the top 5 for evaluation using a unified idea selection module, as discussed below.

**Idea Selection** We conduct a Swiss tournament for idea selection, as previous work (Si et al., 2024) demonstrates that LLMs better assess ideas in pairwise ranking than rating. Over 5 rounds, ideas are paired by similar accumulated scores, with LLMs ranking each pair using the criteria below.

**Idea Evaluation** We assess idea quality using four criteria motivated by previous works (Yang et al., 2024b; Si et al., 2024): significance, novelty, feasibility, and expected effectiveness (abbreviated as exp. effectiveness). These criteria are used in both idea selection and evaluation, with definitions in Appendix D. For automatic evaluation, we conduct tournament ranking and compute ELO scores following Idea Arena (Li et al., 2024a). For human evaluation, we recruit 18 human annotators at the graduate level or above, with academic backgrounds in social science. In addition to the four evaluation criteria, we also ask researchers to assess the overall quality of research ideas.

**Models** We use GPT-4o (`gpt-4o-2024-08-06`) for idea generation and selection, and Gemini-1.5-Pro (Team et al., 2024) (`gemini-1.5-pro-002`) and Claude-3.5-Sonnet (Anthropic, 2024) (`claude-3-5-sonnet-20241022`) serve as judge models to mitigate self-bias.[2] More implementation details, evaluation details, and prompts are in Appendices E, F, and J.

## 4.4 RESULTS

**Automatic Evaluation** Table 2 shows that metadata improves average ratings across all methods, suggesting that incorporating metadata enhances the overall quality of generated research ideas. Expected effectiveness consistently benefits from metadata, with feasibility and significance also improving in most cases, demonstrating metadata's role in generating more empirically grounded and

---

[2]These model versions are used throughout the paper.

impactful ideas. However, novelty declines for AI-Researcher and GPT-Researcher when evaluated by Claude, indicating that data-aware generation may limit highly unconventional ideas.

**Human Evaluation** We perform human evaluation on GPT-Researcher's output, as this method achieves high rankings in the automatic evaluation. For the 50 idea pairs (5 ideas per topic across 10 topics) generated by GPT-Researcher with and without metadata, each pair is annotated by at least two participants, and in total we collected 129 annotations.

As shown in Table 3, the human evaluation results align with the automatic evaluation trends. Incorporating metadata leads to significant improvements in feasibility (20%) and expected effectiveness (18%), though novelty experiences a modest decrease. The overall assessment score increases by 1.5%, suggesting that while metadata strengthens specific quality dimensions, the holistic assessment of research ideas involves balancing multiple criteria. Appendix G demonstrates an example where metadata enhances feasibility and overall quality, and Appendix F analyzes agreement between human evaluators and LLM judges.

## 5 INCORPORATING AUTOMATIC VALIDATION IN IDEA SELECTION

In this section, we first introduce how we conduct preliminary automatic validation and incorporate the validation results into idea selection, as shown in Figure 1 (bottom). We then assess the effectiveness through: (1) a pilot study assessing the hypothesis validation capability, (2) a reference-based evaluation of idea selection performance, and (3) a human evaluation comparing ideas selected with and without the validation process.

### 5.1 INCORPORATING AUTOMATIC VALIDATION INTO IDEA SELECTION

**Feasibility Check** For each idea, we first assess the feasibility of validating its hypotheses. The idea, together with metadata from all available datasets, is provided to an LLM. The model is prompted to determine whether the hypotheses can be tested using the provided datasets and to identify which datasets would be used for the validation.

Specifically, the datasets are indexed numerically. If the model judges the hypotheses to be testable, it outputs the indices of the selected datasets along with a corresponding validation plan. Given the difficulty for LLMs to handle complex data analysis, the number of selected datasets is limited to a maximum of three, with no more than one being a textual dataset.

**Hypothesis Validation** If the idea is deemed testable, the corresponding datasets are then provided to the LLM for validation. We experiment with GPT-4o using the code interpreter assistant, a built-in tool available in GPT models. It achieves superior performance in quantitative reasoning with data (Liu et al., 2024b), while more advanced methods can also be employed in the future. We input the hypotheses along with their corresponding data into the model. The model engages in multi-turn interactions to write and run Python code in a sandbox environment, to validate whether the hypotheses are supported.

**Validation Process Summarization** The raw reasoning traces may be verbose and sometimes contain noise, such as trial-and-error in code execution. To make the output more interpretable and useful for downstream selection, we prompt the LLM to summarize the full validation process into concise natural language steps, including the crucial reasoning process and results that lead to the final conclusion. The summarized validation processes along with the ideas are then provided to the judge model for idea selection.

We use GPT-4o for feasibility check and validation process summarization. Implementation details and prompts are in Appendices E and J, with a case of the validation process shown in Appendix G.

### 5.2 PILOT STUDY: HYPOTHESIS VALIDATION PERFORMANCE ON EXISTING PAPERS

**Experimental Setup** We extract 18 hypotheses from the domain-specific papers collected in §3, of which 10 are supported and 8 are refuted. Hypotheses with insignificant or mixed evidence are excluded. To expand the experimental scope, we sample 50 hypotheses from DiscoveryBench (Majumder et al., 2024b), drawn from 20 papers across diverse fields like humanities, sociology, and

Table 4: Automatic hypothesis validation performancee on existing papers.

(a) Accuracy of the hypothesis validation results.

| Domain | # Papers | # Hypotheses | Accuracy (%) |
|---|---|---|---|
| Diverse | 20 | 100 | 78.0 |
| Climate | 8 | 18 | 72.2 |

(b) Human evaluation of the validation processes.

| Choice | Ratio (%) |
|---|---|
| Mostly Validate the Hypothesis | 50 |
| Partially Validate the Hypothesis | 40 |
| Does not Help Validating | 10 |

(c) Error analysis of the validation processes. Numbers indicate the ratio of validation processes that encounter the error (%).

| Knowledge Recall | Data Analysis | Reasoning Code Generation | Result Interpretation |
|---|---|---|---|
| 30.0 | 63.3 | 30.0 | 33.3 |

economics. Since all these hypotheses are supported in the original papers, we create 50 negative hypotheses by modifying their variables or relations to balance the evaluation dataset. We experiment with GPT-4o using the code interpreter assistant, following the method used to validate hypotheses from generated ideas.

**Results** We begin by evaluating whether the LLM's validation results align with the conclusions presented in the original papers. As shown in Table 4a, the model achieves over 70% accuracy on both general-domain and domain-specific hypotheses. To assess whether this performance stems from memorization, we compare it to a memorization baseline, where the LLM is asked to predict whether the hypotheses are supported without access to the data. Under this setting, the model correctly predicts 65% of DiscoveryBench cases and 55% of climate negotiation cases. Hypothesis validation with data surpasses the memorization baseline by a substantial margin ($\geq$13%), suggesting that the LLM exhibits a meaningful capacity for hypothesis validation.

To evaluate the quality of the validation processes, we conduct a human evaluation, asking two domain experts to review the validation steps for 15 hypotheses drawn from 6 sampled climate negotiation papers, with the annotation interface in Appendix Figure 3. As shown in Table 4b, half of the validation processes mostly support the hypotheses with only minor flaws, while another 40% partially align with the hypotheses but raise significant concerns, such as insufficient control variables. The error analysis in Table 4c reveals that data analysis, particularly involving textual data, is the most challenging aspect for the model. Other common issues, including knowledge recall, reasoning code generation, and result interpretation, also occur in approximately 30% cases.

Despite the imperfections, annotators note that the automatic validation *is helpful as an auxiliary tool for exploratory research*, which aligns well with our intended use of the validation process: as a reference during idea selection.

## 5.3 REFERENCE-BASED AUTOMATIC EVALUATION

Table 5: Accuracy of judge models in ranking ground-truth ideas among LLM-generated ideas (%).

| Judge Model | w. Validation | Significance | Novelty | Feasibility | Exp. Effectiveness | Average |
|---|---|---|---|---|---|---|
| Gemini-1.5-Pro | ✗ | **69.9** | **71.3** | 29.7 | 56.7 | 56.9 |
| | ✓ | 67.3 | 65.8 | **55.6** | **60.6** | **62.3** |
| Claude-3.5-Sonnet | ✗ | **89.4** | 82.5 | 20.1 | 83.8 | 69.0 |
| | ✓ | 88.1 | **86.9** | **46.9** | **93.6** | **78.9** |

We evaluate the impact of validation on idea selection performance, following the setup of Research-Bench (Liu et al., 2025). We prompt LLMs to perform pairwise ranking between ground-truth ideas (extracted from academic papers) and LLM-generated ideas, and compare ranking accuracy with and without access to validation processes.

For each of the 8 climate negotiation papers we collected, we manually extract the research topic and use GPT-Researcher to generate 10 ideas on the same topic, provided with the corresponding dataset description. We then perform automatic validation on both the ground-truth and LLM-generated

Table 6: Human evaluation of ideas selected by Claude-3.5-Sonnet with and without validation processes (%). Significance level is computed with McNemar' test (* for $p < 0.1$, ** for $p < 0.05$, and *** for $p < 0.01$).

|  | w. Validation | Tie | w/o Validation | Significance Level |
|---|---|---|---|---|
| Significance | **37.5** | 27.5 | 35.0 | - |
| Novelty | **45.0** | 21.7 | 33.3 | - |
| Feasibility | **40.0** | 33.3 | 26.7 | * |
| Exp. Effectiveness | **43.3** | 27.5 | 29.2 | * |
| Overall | **42.5** | 21.7 | 35.8 | - |

ideas, and ask judge LLMs to pairwise compare the ground-truth ideas with the generated ideas under the same topic. Accuracy is defined as the proportion of comparisons where the ground-truth idea is ranked higher. To mitigate position bias, each pair is evaluated twice with reversed positions, and the results are averaged.

We use Gemini-1.5-Pro and Claude-3.5-Sonnet as judge LLMs, and results are shown in Table 5. For both models, incorporating validation leads to consistently higher average ranking accuracy compared to without validation. Improvements are particularly notable in feasibility and expected effectiveness, while a slight decrease is observed in the judgment of significance and novelty.

### 5.4 HUMAN EVALUATION

We then conduct a human evaluation comparing LLM-generated ideas selected with and without validation processes. For the 50 ideas generated by GPT-Researcher in §4 on each research topic, we use Claude-3.5-Sonnet, which performs better in the reference-based evaluation, to select the top 5 ideas in two settings: (1) based on the idea content alone, and (2) based on both the idea and its validation process. Human annotators then perform pairwise evaluations of the two sets, using the same evaluation setup as described in §4. Each pair is annotated by at least two participants, and in total we collected 120 annotations.

As shown in Table 6, ideas selected with validation processes are ranked higher across all dimensions, with the largest improvements observed in feasibility and expected effectiveness. This aligns with the reference-based evaluation results and suggests that validation processes provide a valuable signal for enhancing idea selection.

## 6 HUMAN STUDY: ARE THE LLM GENERATED IDEAS INSPIRING TO RESEARCHERS?

Beyond evaluating idea quality, we are interested in whether LLM-generated ideas can be useful in real-world academic settings. We conduct a human study to investigate whether ideas generated by LLMs, along with related data and validation processes, can inspire researchers to formulate their own research ideas.

### 6.1 EXPERIMENT DESIGN

We recruit 23 participants from a social science course to take part in the study. Among them, 19 are undergraduate or graduate students, and the remaining 4 are more senior researchers holding a PhD in a related field. Participants are presented with four research topics related to climate negotiations and asked to select two topics they are personally interested in. For each selected topic, they are asked to propose a research idea.

For one of the two topics, participants are provided with three reference ideas, accompanied by data snippets used in automatic validation and the validation processes. The reference ideas are from the experiment of §5.4, generated by GPT-Researcher with metadata and selected by Claude-3.5-Sonnet based on validation processes. For each idea, we present the first 10 lines of the datasets used during validation. Both the raw validation traces and the summarized versions are provided, and participants may choose which format to consult. For the other topic, participants are not given

any references. They are allowed to browse the Internet and search for literature but are not permitted to use LLMs. Additional details including the experiment interface are provided in Appendix F.

## 6.2 RESULTS

Table 7: Human study results on the inspirational value of LLM-generated ideas (%).

(a) Human comparison of ideas proposed by participants with vs. without access to references (including LLM-generated ideas, related data and validation processes).

| | w. Reference | Tie | w/o Reference |
|---|---|---|---|
| Significance | **39.1** | 21.7 | **39.1** |
| Novelty | **43.5** | 23.9 | 32.6 |
| Feasibility | **50.0** | 17.4 | 32.6 |
| Exp. Effectiveness | **39.1** | 28.3 | 32.6 |
| Overall | **39.1** | 28.3 | 32.6 |

(b) Participant feedback on the helpfulness of reference ideas, data segments, and validation processes for generating their own research ideas. High and medium helpfulness correspond to the `very helpful` and `somewhat helpful` options, respectively.

| | High | Medium | Not Helpful |
|---|---|---|---|
| Reference Ideas | 61.1 | 33.3 | 5.6 |
| Data Segments | 33.3 | 50.0 | 16.7 |
| Validation Processes | 55.5 | 38.9 | 5.6 |

**Quality of Ideas** We ask human experts to evaluate participant-proposed ideas using the same evaluation setup as §4. Since the number of ideas proposed with and without references may differ for a given topic, we first pair ideas from both settings one-to-one. For any excess ideas in one setting, we randomly sample additional ideas from the other to complete the set of pairs.

As shown in Table 7a, ideas proposed with references demonstrate higher overall quality. Specifically, improvements are observed in novelty, feasibility, and expected effectiveness.

**Feedback from Participants** To understand whether participants find the references helpful and how they use them, we collect self-reported feedback. Participants are asked to rate the helpfulness of the reference ideas, data segments, and validation processes separately, using a three-point scale: `very helpful`, `somewhat helpful`, and `not helpful`.

As shown in Table 7b, all three components are generally found helpful by most participants. Reference ideas and validation processes are rated as `very helpful` by more than half of the participants. The data segments receive relatively lower ratings, likely because raw data often requires additional interpretation or context to be fully understood, whereas ideas and validation outputs provide more immediately actionable guidance.

Several participants provide detailed feedback. One student notes *they build their own idea by extending the most interesting reference idea*, while another mentions that *the concepts and measurements in the references help refine their own research direction*. A professor also remarks that *the references served as useful shortcuts and they can revise upon them*. These insights highlight how LLM-generated references support researchers based on their background and research stage.

## 7 CONCLUSION AND DISCUSSION

Facing the challenge of generating feasible and effective research ideas with LLMs, we propose a framework that incorporates data into research ideation through metadata and automatic validation. Experiments show that by guiding idea generation with dataset descriptions and selecting ideas given automatic validation processes, LLMs are able to propose ideas that are more feasible and more likely to be effective. Beyond quality improvements, we find that these LLM-generated ideas, along with their validation traces, can serve as valuable inspiration for human researchers.

In discussing this work with social sciences researchers, we encounter thoughtful reflections on the value of LLM-generated ideas. Some researchers question whether ideas proposed by LLMs truly matter if they do not originate from human "care" or intention. These conversations raise deeper questions about the nature of research: What distinguishes a good idea from a valuable idea? How could LLM-generated ideas contribute to real-world research in ways that augment human creativity? While we provide a preliminary case study of such use in §6, these questions remain open and worth future exploration.

ETHICS STATEMENT

Research ideas generated by LLMs may reflect biases present in their training data and could unintentionally resemble existing work without proper citation. Therefore, these ideas should not be adopted for practical use without thorough validation. Furthermore, any use of LLM-generated ideas should be disclosed transparently to ensure ethical integrity.

REPRODUCIBILITY STATEMENT

We provide the data in the supplementary material, and will release it to the public.

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

## A  DATA COLLECTION

Table 8: List of datasets and the corresponding data descriptions in CLIMATEDATABANK.

| Name | Description |
|------|-------------|
| **Textual Data** | |
| National communications | National communications submitted by countries every four years (Annex I Parties) or eight years (Non-Annex I Parties), outlining their efforts to address climate change. |
| High-level statements | High-level climate change conference speeches, covering the formal statements made by country-representatives at COPs (2010-2023). |
| Earth negotiation bulletins | Reports summarizing the negotiation process and main outputs of UNFCCC meetings, including both daily reports and summary reports (1995-2024). |
| Business statements | UNFCCC statements of business associations in the span of eight years (2007-2014). |
| **Panel Data** | |
| Meeting attendance records | Attendee records from all UNFCCC COP meetings (1995-2023), including their delegation, job, gender, and so on. |
| GDP | The sum of gross value added by all resident producers in the economy plus any product taxes and minus any subsidies not included in the value of the products (in US$). |
| GDP per capita | Gross domestic product divided by midyear population (in US$). |
| Population | The population of the country, which counts all residents regardless of legal status or citizenship. |
| Foreign direct investment | Direct investment equity flows in the reporting economy, which is the sum of equity capital, reinvestment of earnings, and other capital (in US$). |
| Life expectancy at birth | The number of years a newborn infant would live if prevailing patterns of mortality at the time of its birth were to stay the same throughout its life. |
| Gender parity index | The ratio of girls to boys enrolled at primary and secondary levels in public and private schools. |
| CO2 emissions per capita | Carbon dioxide ($CO_2$) emissions excluding LULUCF per capita. |
| Forest area | Land under natural or planted stands of trees of at least 5 meters in situ (sq. km). |
| Natural resources rent | Rents from coal, oil and natural gas production (% of GDP). |
| Trade openness index | Sum of exports and imports of goods and services, divided by gross domestic product, expressed as a percentage. |
| Democracy index | The country's level of democracy, ranging from -10 to 10 (fully democratic). |
| World risk index | Higher scores indicate higher vulnerability to climate change. |
| ND-GAIN vulnerability index | Higher scores indicate higher vulnerability to climate change. |
| **Cross-Sectional Data (in 2025)** | |
| Member of AOSIS | Whether the country is a member of the Alliance of Small Island States (AOSIS). |
| Member of OPEC | Whether the country is a member of the Organization of the Petroleum Exporting Countries (OPEC). |
| Member of G20 | Whether the country is a member of G20. |
| Annex I country | Whether the country is an Annex I country. |

Table 8 presents the full list of datasets in CLIMATEDATABANK and corresponding data descriptions. Table 9 demonstrates the climate negotiation papers we collect for the automatic validation experiments.

National communications, high-level statements, and business statements are collected from the UNFCCC website[3], which allows free download and copy. Earth negotiation bulletins are collected from the ENB website[4]. Meeting attendance records are from Blinova et al. (2024) under the CC0 1.0 license. Democracy index is from Marshall et al. (2014). World risk index is from Welle &

---

[3]https://unfccc.int/
[4]https://enb.iisd.org/

Table 9: List of reference papers used in automatic validation experiments.

| ID | Paper Title |
|---|---|
| 1 | The Multifaceted Nature of Global Climate Change Negotiations (Bagozzi, 2015) |
| 2 | A Closer Look at the Information Provision Rationale: Civil Society Participation in States' Delegations at the UNFCCC (Böhmelt, 2013) |
| 3 | Sectors, Pollution, and Trade: How Industrial Interests Shape Domestic Positions on Global Climate Agreements (Genovese, 2019) |
| 4 | The domestic politics of international climate commitments: which factors explain cross-country variation in NDC ambition? (Tørstad et al., 2020) |
| 5 | Which Countries Send More Delegates to Climate Change Conferences? Analysis of UNFCCC COPs, 1995–2015 (Kaya & Schofield, 2020) |
| 6 | The Institutionalization of a Cleavage: How Differential Treatment Affects State Behavior in the Climate Negotiations (Castro & Kammerer, 2021) |
| 7 | How Do Countries Frame Climate Change? A Global Comparison of Adaptation and Mitigation in UNFCCC National Communications (Wright et al., 2023) |
| 8 | Institutional Roots of International Alliances: Party Groupings and Position Similarity at Global Climate Negotiations (Genovese et al., 2023) |

Birkmann (2015) under the CC BY license. ND-GAIN vulnerability index is from Chen et al. (2015). All other data in CLIMATEDATABANK are from World Bank Open Data under the CC-BY 4.0 license.

## B  RESEARCH TOPICS

We generate 10 climate negotiation-related research topics using GPT-4o, with the prompt: *Could you propose 10 research topics related to climate negotiation? The topics should be important for social science researchers, like in the community of political science and climate policies. The output should be in JSON format, with the key being the topic name and the value being the explanation. Each topic name should be within five words.*

The created topics are listed in Table 10, and their quality has been reviewed and verified by human experts. All the research topics and generated ideas throughout this paper are in English.

## C  RESEARCH IDEA GENERATION METHODS

We experiment with three prevalent research idea generation methods in §4:

- AI-Researcher (Si et al., 2024): This method first retrieves papers related to the given research topic from Semantic Scholar, uses the retrieved papers to ground idea generation, produces a large number of candidate ideas, and then ranks them to identify the best ones.

- GPT-Researcher (Elovic, 2023): This method builds a multi-agent framework consisting of planner, executor, and publisher agents. The planner generates plans, while the executor gathers relevant information. The publisher aggregates all information and generates the research ideas.

- Chain-of-Ideas (Li et al., 2024a): This method enhances the literature search module by organizing relevant literature in a chain structure to effectively mirror the progressive research development.

To ensure a fair comparison, each method is uniformly tasked with generating 50 candidate ideas for each research topic. We then use the same idea selection module to rank and select the top ideas.

## D  EVALUATION CRITERIA

The ideas are evaluated according to the following four criteria:

Table 10: Climate negotiation research topics used in this paper.

| Topic | Description |
|---|---|
| Adaptation vs. Mitigation Focus | Study the negotiation dynamics and policy priorities between adaptation and mitigation efforts, and the factors influencing their prominence in different countries' strategies. |
| Climate Finance Politics | Examine the political challenges and negotiations around climate finance, including funding commitments, allocation mechanisms, and equity in financial support for adaptation and mitigation. |
| Climate Justice and Equity | Investigate how principles of justice and equity are integrated into climate negotiations and their impacts on policy outcomes for different countries and communities. |
| Compliance and Monitoring Mechanisms | Focus on the systems in place for ensuring adherence to international climate agreements, and the effectiveness of these mechanisms in promoting accountability. |
| Impacts of Domestic Policies | Explore how domestic climate policies of influential nations affect their negotiation positions and the overall dynamics in international climate agreements. |
| Historical Responsibility Debates | Analyze discussions around historical responsibility for climate change and how these debates shape fairness principles and burden-sharing in negotiations. |
| Negotiation Strategies and Tactics | Analyze the negotiation strategies employed by countries or blocs in climate negotiations, including coalition-building, bargaining tactics, and compromise-making. |
| Role of Non-State Actors | Study the influence and participation of non-state actors, such as NGOs, private sector, and indigenous groups, in shaping climate negotiation agendas and outcomes. |
| Power Dynamics and Influence | Examine the roles of different countries, especially major emitters versus vulnerable states, and their influence in shaping international climate agreements and commitments. |
| Technology Transfer and Collaboration | Explore the negotiations related to technology transfer, the barriers to effective collaboration, and how they impact developing countries' abilities to meet climate goals. |

- Significance: Whether the research idea is impactful to the researchers and the broader public.

- Novelty: Whether the idea contributes fresh insights and perspectives to the existing body of knowledge.

- Feasibility: Whether the study can be done with available resources, time, and technology, typically within a one-year scope for a political science PhD student.

- Expected Effectiveness: How likely the proposed idea will successfully achieve its intended outcomes, i.e., how likely the theory will be supported by empirical evidence.

A more detailed version of the criteria is shown in Table 11. This is provided to LLMs during idea selection and automatic evaluation, as well as to human annotators for reference.

## E  IMPLEMENTATION DETAILS

For the research idea generation methods, we adhere to their original hyperparameters but modify the idea generation prompts to include instructions related to idea formats, and add the metadata. Since in social science research, policy implications are frequently invoked to demonstrate a study's broader relevance and impact, we also ask LLMs to explain the policy implications of generated ideas in the idea generation step. Note that this is only for self-awareness and is excluded from subsequent idea selection and evaluation.

For idea selection in §4, we follow AI-Researcher's tournament ranking method but adapt it by having the model rank idea pairs based on the four evaluation aspects separately. The idea that wins

Table 11: Detailed criteria for evaluating ideas.

**Significance**
1. Impact on the Field:
- Does the research have the potential to influence future work in the field significantly?
- Will it change the way scholars and practitioners think about a particular issue or problem?
2. Relevance to Current Problems:
- Does the research tackle urgent or pressing issues faced by society today?
- How does it contribute to solving real-world problems or advancing public policy?
3. Advancement of Theoretical or Practical Understanding:
- Does it deepen our theoretical insights or provide new frameworks for understanding?
- Can the findings be translated into practical applications or technologies that benefit society?
**Novelty**
1. Originality:
- Is the research question unique or a significant departure from existing studies?
- Does the theory offer a new perspective or challenge prevailing paradigms?
2. Innovation in Approach:
- Are there novel methodologies or analytical techniques proposed?
- Does it introduce new datasets or sources of evidence?
3. Contribution to Knowledge:
- Does the idea fill a significant gap in the literature?
- How does it expand or refine existing theories or models?
**Feasibility**
1. Resource Availability:
- Can the necessary data or materials be accessed or acquired with reasonable effort?
- Are funding, human resources, and technical support sufficient?
2. Timeline Appropriateness:
- Can the study be realistically completed within one year?
- Does the research have clear stages with achievable milestones?
3. Technical and Methodological Soundness:
- Are the proposed methodologies practical and well-founded?
**Expected Effectiveness**
1. Theoretical Rigor:
- Is the theory logically sound with well-defined constructs and relationships?
- How well are the hypotheses grounded in existing literature and theory?
2. Empirical Evidence Potential:
- How robust is the potential for empirical evidence to support the theory?
- Are the proposed indicators measurable and likely to yield clear data?

in more aspects is considered the winner, and a tie occurs if the two ideas win an equal number of aspects.

The temperature is set to 0 for all steps after idea generation. The maximum number of output tokens is set to 1024 for the feasibility check, idea selection, and automatic evaluation. Experiments are conducted on 8 NVIDIA A800 GPUs.

# F  EVALUATION DETAILS

## F.1  AGREEMENT ANALYSIS

We analyze the agreement between human evaluators and automatic judge models in evaluating LLM-generated ideas.

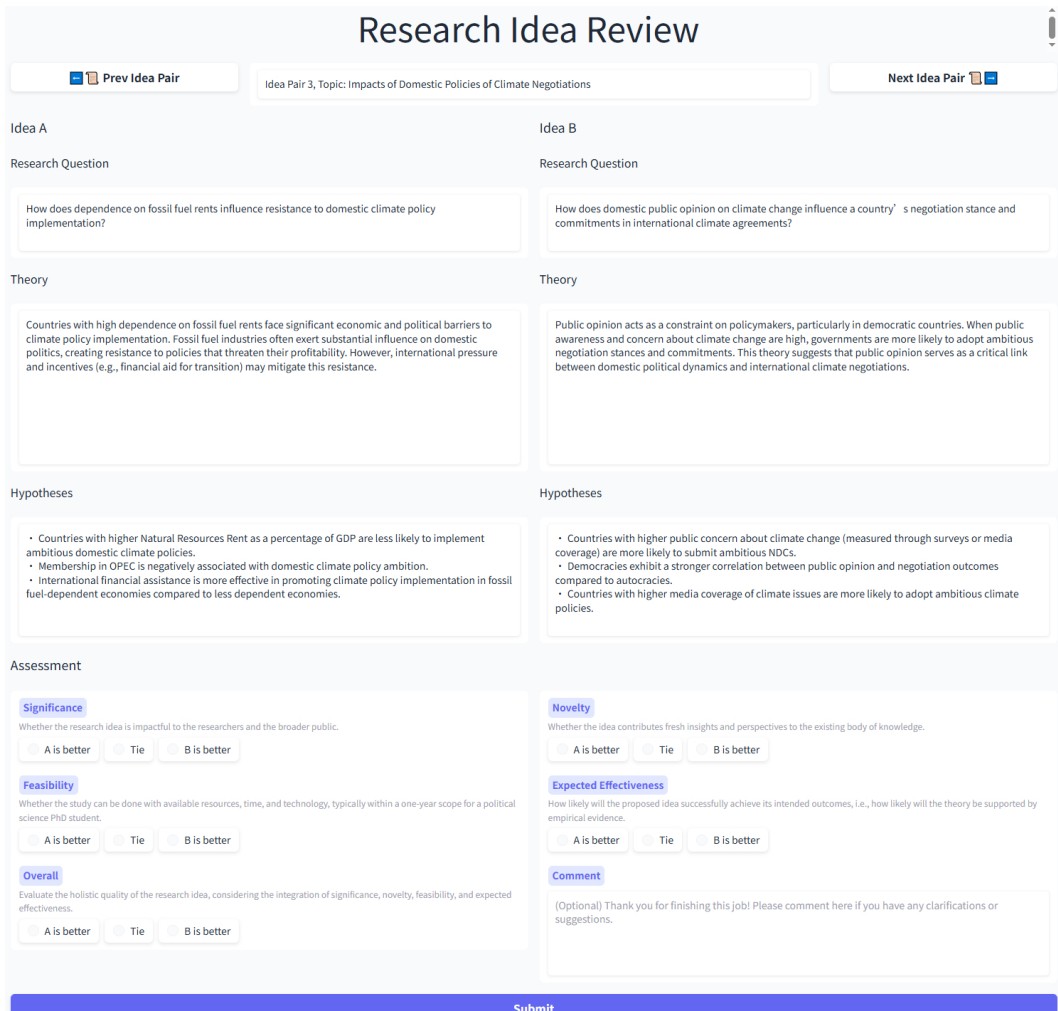

Figure 2: Annotation interface for human evaluation of idea pairs.

For human annotators, we measure inter-rater agreement using Krippendorff's alpha, given that annotator assignments vary across idea pairs. The average Krippendorff's alpha obtained is 0.30, which is much better than a random rating but far from perfect agreement. This moderate agreement reflects the inherent subjectivity in assessing the quality of research ideas, which is also observed in prior research (Si et al., 2024).

Regarding the two LLM judges in §4, their average Krippendorff's alpha is 0.24, slightly lower than that observed for human evaluators. However, when we aggregate their individual predictions and computed ELO scores, the Pearson correlation between their generated ELO scores is 0.67. This demonstrates a moderate to high correlation between the two LLM judges' overall assessments, suggesting a consistent ranking capability despite some individual disagreement on specific idea pairs.

## F.2 ANNOTATION INTERFACES

Figures 2 and 3 show the annotation interfaces for human evaluations of idea pairs (in §4 - §6) and hypothesis validation processes (in §5.2), respectively. All annotators are fairly paid with more than $10 per hour.

In the human study of §6, participants are given 20 minutes to propose one research idea for each research topic they select. The experiment interface for this task is shown in Figure 4.

## G CASE STUDY

Table 12: Examples of ideas generated by GPT-Researcher with and without metadata.

| Topic | Compliance and Monitoring Mechanisms |
|---|---|
| **Idea 1: Generated without Metadata** | |
| **Research Question** | How does the Global Stocktake process under the Paris Agreement influence collective progress toward climate goals, and what factors enhance its effectiveness? |
| **Theory** | The theory posits that the Global Stocktake serves as a critical feedback mechanism that fosters collective action by identifying gaps, sharing best practices, and increasing ambition. Its effectiveness depends on the inclusivity of the process, the quality of data used, and the willingness of states to act on its findings. The theory also suggests that the Global Stocktake is more effective when it incorporates inputs from non-party stakeholders and aligns with domestic political priorities. |
| **Hypotheses** | 1. The inclusivity of the Global Stocktake process (e.g., participation of non-party stakeholders) positively correlates with its effectiveness in fostering collective action. 

 2. High-quality data used in the Global Stocktake increases its impact on state ambition and compliance. 

 3. States that align their domestic policies with Global Stocktake findings are more likely to achieve their climate goals. 

 4. The Global Stocktake is less effective in fostering collective action when major emitters (e.g., the US, China) fail to participate meaningfully. |
| **Idea 2: Generated with Metadata** | |
| **Research Question** | How do domestic political institutions influence the effectiveness of compliance monitoring in international climate negotiations? |
| **Theory** | Domestic political institutions play a critical role in shaping a country's commitment and ability to comply with international climate agreements. Democratic institutions, with their emphasis on transparency and accountability, are theorized to facilitate better compliance monitoring. Conversely, autocratic regimes may prioritize short-term economic gains over long-term climate commitments, leading to weaker compliance. Additionally, institutional mechanisms such as independent regulatory agencies, judicial oversight, and civil society engagement may enhance the credibility and accuracy of compliance monitoring. |
| **Hypotheses** | 1. Countries with higher Democracy Index scores are more likely to submit accurate and timely compliance reports to international climate bodies. 

 2. The presence of independent regulatory agencies positively correlates with the quality of compliance monitoring in climate negotiations. 

 3. Civil society engagement, as measured by the number of environmental NGOs per capita, enhances the accuracy of compliance reporting. 

 4. Autocratic regimes are more likely to underreport their emissions compared to democratic regimes, controlling for economic and environmental factors. |

**Ideas generated with and without metadata**   Table 12 presents an example of ideas generated by GPT-Researcher under the same topic. Idea 1, generated without metadata, contains undefined terms such as inclusivity and high-quality data, while Idea 2, which is guided by metadata, introduces clear and measurable hypotheses. The integration of metadata makes the research idea more actionable, increasing the likelihood of meaningful findings and improving overall quality.

**The automatic validation process**   Table 13 showcases an example of the automatic validation process. Based on an idea generated by GPT-Researcher under the topic *Role of Non-State Actors*, the LLM first conducts a feasibility check and selects three datasets from the CLIMATEDATABANK. It then performs hypothesis validation and summarizes the validation process in natural language.

**Top-ranked and bottom-ranked ideas selected with access to the automatic validation process**
In the idea selection stage, the idea in Table 13 (referred to as Idea A) is ranked the first among ideas related to *Role of Non-State Actors*. And Table 14 shows the bottom-ranked idea (referred to as Idea B) under the same topic.

Table 13: Examples of the automatic validation process.

| Topic | Role of Non-State Actors |
| --- | --- |
| **Idea Generated** | |
| **Research Question** | How do non-state actors influence the ambition levels of national climate commitments under the Paris Agreement? |
| **Theory** | Non-state actors (NSAs), such as businesses, civil society organizations (CSOs), and research institutions, play a critical role in driving climate ambition by pressuring governments to adopt more stringent climate policies. This influence stems from their ability to mobilize public opinion, provide technical expertise, and create accountability mechanisms. The theory posits that NSAs are particularly effective in democracies, where governments are more responsive to public pressure, and in countries with high trade openness, where businesses are incentivized to align with international climate norms to maintain competitiveness. |
| **Hypotheses** | 1. Countries with higher levels of NSA participation in UNFCCC meetings will exhibit greater increases in the ambition of their Nationally Determined Contributions (NDCs) over time. |
| | 2. The impact of NSA participation on NDC ambition will be stronger in democracies compared to autocracies. |
| | 3. Trade openness moderates the relationship between NSA participation and NDC ambition, with more open economies showing a stronger positive relationship. |
| **Feasibility Check** | |
| **Feasibility** | Yes |
| **Data Selected** | 1 (national communications), 5 (meeting attendance records), 16 (democracy index) |
| **Hypothesis Validation** | |
| (raw traces omitted due to space limit) | |
| **Validation Process Summarization** | |

1. The datasets were successfully loaded, containing information on national communications, attendance at UNFCCC meetings, and democracy index scores. The approach involves testing three hypotheses: 1) NSA attendance correlates with increased NDC ambition, 2) this relationship differs between democratic and autocratic countries, and 3) trade openness influences the relationship between NSA participation and NDC ambition.

2. NSA participation was quantified by counting attendees per country and year. Ambition levels in national communications were inferred by counting occurrences of keywords like 'increase', 'commitment', and 'target'.

3. Ambition scores were calculated based on keyword occurrences in national communications. The data was integrated with NSA participation and democracy index data, aligning them by country and year.

4. The integrated dataset was prepared, containing NSA participation, ambition scores, and democracy index scores. This dataset is ready for hypothesis testing.

5. Hypothesis 1 was tested, showing a statistically significant positive correlation between NSA participation and NDC ambition scores, supporting the hypothesis.

6. Hypothesis 1 is supported, indicating a positive correlation between NSA participation and NDC ambition.

7. Hypothesis 2 was tested by segmenting data into democracies and autocracies based on democracy index scores. The relationship was not significantly stronger in democracies, so the hypothesis is not supported.

8. Hypothesis 2 is not supported, as the relationship between NSA participation and NDC ambition is not significantly stronger in democracies compared to autocracies.

9. Hypothesis 3 could not be tested due to the absence of trade openness data. The final results are: Hypothesis 1 is supported, Hypothesis 2 is not supported, and Hypothesis 3 needs more data.

While both ideas offer highly significant research avenues, the ability to empirically measure "NDC ambition" and "NSA participation in UNFCCC meetings" makes Idea A's hypotheses more directly testable. In contrast, Idea B's focus on "perceived legitimacy" and "public support" for international agreements presents greater methodological challenges. Measuring these concepts often requires extensive surveys or complex qualitative methods, which can introduce variability and impact the rigor of the findings.

Table 14: Example of the bottom-ranked idea under the topic *Role of Non-State Actors*, comparing to the top-ranked idea shown in Table 13.

| Topic | Role of Non-State Actors |
|---|---|
| **Idea with the Lowest Ranking** | |
| **Research Question** | How do non-state actors contribute to the perceived legitimacy of international climate agreements, and what factors enhance their effectiveness in this role? |
| **Theory** | The legitimacy of climate agreements depends not only on state actors but also on the active participation of NSAs, which represent diverse stakeholder interests. NSAs enhance legitimacy by advocating for justice, inclusivity, and transparency in climate negotiations. Their effectiveness in this role is influenced by their organizational capacity, networks, and alignment with global norms. |
| **Hypotheses** | 1. NSAs with higher levels of organizational capacity (e.g., funding, staff) are more effective in enhancing the legitimacy of climate agreements. |
| | 2. NSAs that engage in transnational networks are more likely to influence public perceptions of climate agreements. |
| | 3. The inclusion of NSAs in formal negotiation processes positively correlates with higher public support for climate agreements. |

## H   THE USE OF LLMS

We use LLMs in polishing the writing, but LLMs do not play a significant role like research ideation or writing the content directly.

## I   LIMITATIONS

**Task Scope**   While our experiments focus on topics related to climate negotiations, the proposed method could be applied to other quantitative social science research areas. We believe that incorporating data could also enhance the generation of research ideas in other domains, such as computer science, but this would need further development of the method.

**Exploration of LLMs and Validation Methods**   Due to the high cost of human evaluation, our experiments focus on a single LLM and a specific automatic validation method. Future studies could systematically evaluate how different models and validation methods impact idea quality.

**Trade-off between Novelty and Feasibility**   The introduction of metadata improves feasibility but leads to a modest decline in novelty. This suggests that although LLMs are not explicitly restricted to the provided data, the metadata implicitly narrows their scope of imagination. Future works could broaden the data scope from existing data to data that can be collected, or better integrate literature with data to maintain a balance between creativity and feasibility.

## J   PROMPTS

Table 15 presents the prompt for idea generation using AI-Researcher. The same instructions regarding idea format, example ideas, and metadata are provided to GPT-Researcher and Chain-of-Ideas. The example ideas are drawn from existing academic papers.

Table 16 shows the prompt used for both idea selection and automatic evaluation in §4. The difference is that idea selection is conducted by the LLM for idea generation, whereas in automatic evaluation, other LLMs are used to reduce bias.

Tables 17 through 19 display the prompts for the automatic validation process, including feasibility checks, hypothesis validation, and validation process summarization. Table 20 outlines the prompt for idea selection in §5, which differs from Table 16 by incorporating the validation process.

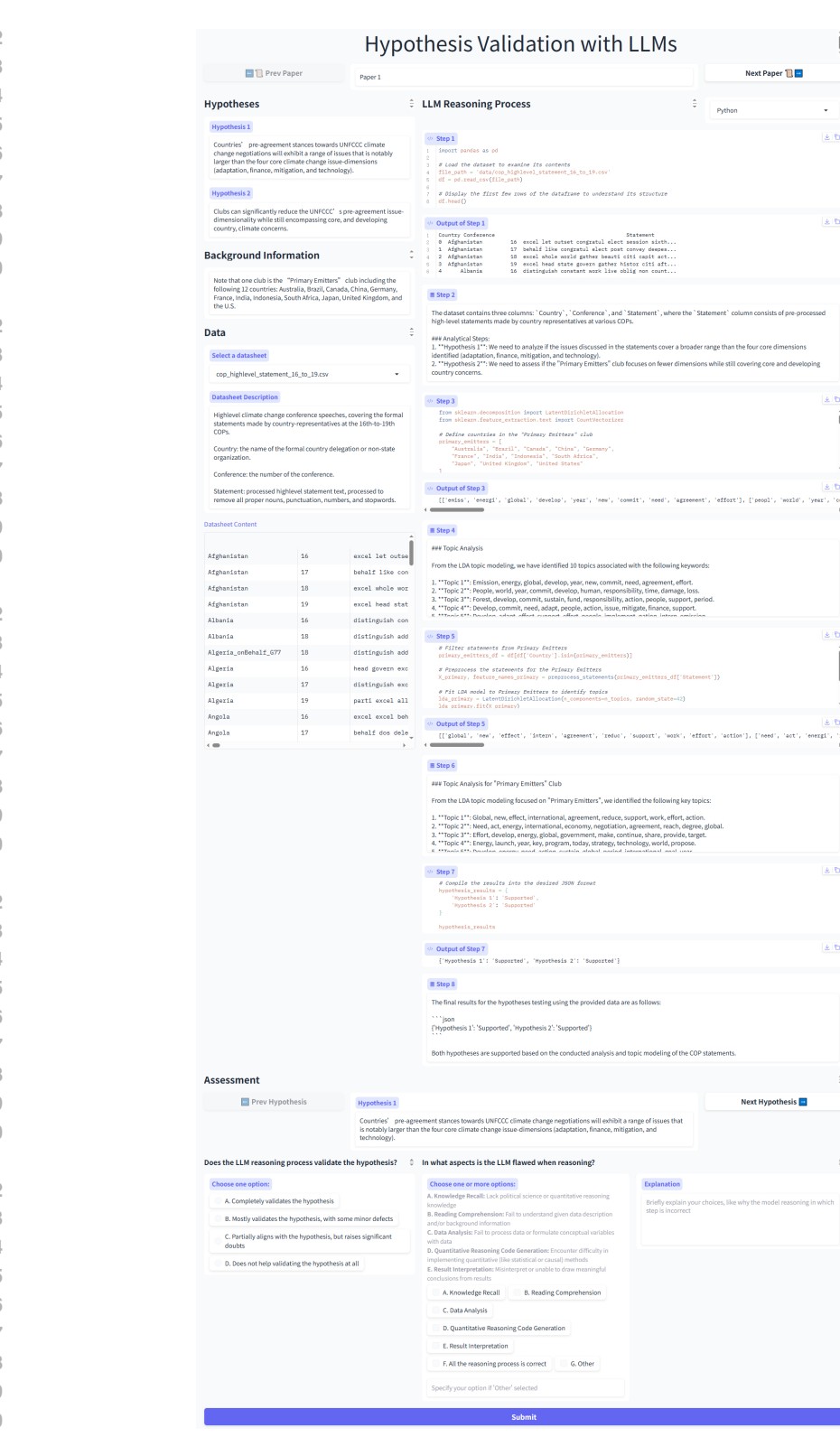

Figure 3: Annotation interface for human evaluation of hypothesis validation processes.

## Please submit your research idea

Topic 1    Topic 2

Topic 1

**Technology Transfer and Collaboration of Climate Negotiations**

Reference Idea 1    Reference Idea 2    Reference Idea 3

Content of Idea 1

**Research Question**

How do participation in carbon markets and the adoption of Article 6 mechanisms under the Paris Agreement influence technology transfer agreements in climate negotiations?

**Theory**

Carbon markets create economic incentives for countries to engage in cooperative climate action, including technology transfer. The operationalization of Article 6 mechanisms under the Paris Agreement provides a framework for international collaboration, enabling countries to trade carbon credits while integrating technology transfer as part of their mitigation strategies. Countries that actively participate in carbon markets may be more likely to secure technology transfer agreements as part of their broader climate commitments.

**Hypotheses**

· Countries that actively participate in carbon markets are more likely to secure technology transfer agreements during climate negotiations.
· The positive effect of carbon market participation on technology transfer is stronger for countries with lower CO2 emissions per capita, as they are seen as more credible partners.
· The operationalization of Article 6 mechanisms positively moderates the relationship between carbon market participation and technology transfer agreements.

[ Show Data ]    [ Show Code ]

Content of Your Idea

**Research Question**

Please type in the research question...

**Theory**

Please type in the theory...

**Hypotheses**

Please type in the hypotheses...

Feedback

**Are the reference ideas helpful in creating your research idea?**

○ Very helpful    ○ Somewhat helpful    ○ Not helpful

**Is the data helpful in creating your research idea?**

○ Very helpful    ○ Somewhat helpful    ○ Not helpful

**Is the validation process helpful in creating your research idea?**

○ Very helpful    ○ Somewhat helpful    ○ Not helpful

**Share your comment**

Both positive and negative comments are welcomed

Total time remaining: 39:46

**Submit**

Figure 4: Experiment interface for the human study of proposing research ideas.

---

**Prompt for Idea Generation**

You are an expert researcher in political science. Now I want you to help me brainstorm some new research ideas on the topic of {research topic}.

Here are some relevant papers on this topic just for your background knowledge:

{titles and abstracts of related literature}

The above papers are only for inspiration and you should not cite them and just make some incremental modifications. Instead, you should make sure your ideas are novel and distinct from the prior literature.

*Here are existing data related to this topic:*
*Textual data:*
*1. National communications: National communications submitted by countries every four years (Annex I Parties) or eight years (Non-Annex I Parties), outlining their efforts to address climate change.*
*2. High-level statements: ...[omitted]...*
*Panel data:*
*5. Meeting attendance records: ...[omitted]...*
*Cross-sectional data:*
*19. Member of AOSIS: ...[omitted]...*

You should generate {number of ideas to generate} different ideas on this topic. Try to be creative and diverse in idea generation, and do not repeat any similar ideas.

You should aim for research that can be published in top political science journals. Good research should contribute to theoretical value and/or policy implications.

Each idea should be described as:

(1) Research Question: Clearly propose a research question, which should be closely related to the topic. Research questions can delve into issues of what, why, how, when, and so forth. Interesting research questions are those that intellectually appeal to political scientists, address concerns of a broad population and decision makers, and where the answers are not obvious.

(2) Theory: Develop a theory that reasonably speculates on the answer to the research question, including a statement about why the proposed answer is correct. A theory is a system of concepts and relationships between those concepts, that collectively presents a logical, systematic, and coherent explanation of a phenomenon of interest.

(3) Hypotheses: Propose 1-5 hypotheses derived from the theory. The hypotheses identify observable implications of the theory, i.e., things we would observe if the theory is correct, and make predictions about relationships between measurable indicators of the theory's concepts.

(4) Policy Implication: Explain how the research could help policymakers to adjust their decisions, or implement policy more effectively or justly.

Here are examples of research ideas on other topics.

{content of two example ideas}

You should make sure to come up with your own novel and different ideas for the specified topic: {research topic}. You should make each idea standalone and not dependent on the other ideas.

You should avoid repeating generating ideas with the following existing research questions, and try to be different and diverse:

{existing ideas generated}

Please write down your {number of ideas to generate} ideas. Output the ideas in json format as a dictionary, where the key is 'ideas', and the value is a list of ideas. Each idea has keys 'Research Question', 'Theory', 'Hypotheses', and 'Policy Implication'. The value of 'Hypotheses' is a list of strings, and the value of other keys is a string.

Table 15: Example prompt for idea generation with AI-Researcher. The same idea format instructions, example ideas, and metadata are also provided to GPT-Researcher and Chain-of-Ideas.

**Prompt for Both Idea Selection and Automatic Evaluation in §4**

You are an expert researcher in political science. You are given two research ideas related to the topic {research topic}. Your task is to identify which idea is better from the following four dimensions 'Significance', 'Novelty', 'Feasibility', and 'Expected Effectiveness'.

Each research idea comprises the following three parts.

Research Question: A specific question about a behavior, event, or phenomenon of interest that the researcher wishes to seek answers for in the research.

Theory: Reasonably speculate on the answer to the research question, including a statement about why the proposed answer is correct.

Hypotheses: Identify observable implications of the theory, i.e., things we would observe if the theory is correct, and make predictions about relationships between measurable indicators of the theory's concepts.

Evaluation Criteria:

{detailed content of the evaluation criteria}

Note: Please make your decision based on the weighted assessment of sub-criteria to avoid subjective bias. Avoid any position biases and ensure that the order of the two ideas does not influence your decision. DO NOT allow the LENGTH of the ideas to influence your evaluation. Be as objective as possible.

Here are the two research ideas for you to assess:

Idea 1:

{content of idea 1}

Idea 2:

{content of idea 2}

Please provide an explanation supporting your assessment. At the last line of your response, format your assessment in JSON with the keys: 'Significance', 'Novelty', 'Feasibility', and 'Expected Effectiveness'. The value of each key is an integer ranging from 0 to 2. 0 means a tie, 1 means idea 1 is better, and 2 means idea 2 is better.

Table 16: Example prompt for both idea selection and automatic evaluation in §4.

**Prompt for Feasibility Check**

You are an expert researcher in political science. Given a research idea with the components of 'Research Question', 'Theory', 'Hypotheses', along with descriptions of existing data, please determine the feasibility of validating the hypotheses using the provided data.

Here is the research idea:
{content of the idea}
Here is the existing data:
{content of the metadata in CLIMATEDATABANK }

Your task is as follows:
1. Feasibility Assessment:
- Evaluate whether it is possible to validate the hypotheses with the given data.
- If feasible, provide a validation plan and specify the data that will be used by their numbers. A hypothesis is considered feasible to validate if the concepts in the hypothesis can be measured with existing data.
- If not feasible, output 'Feasibility' as 'No'. Note that the theory provides an answer and explanation to the research question, and the hypotheses identify observable implications of the theory.
2. Output Requirements:
- Format your response in JSON with the keys: 'Feasibility', 'Validation Plan', and 'Data Used'.
- 'Feasibility': This can take values from ['Yes', 'No']. It indicates whether the hypotheses can be validated with the existing data.
- 'Validation Plan': A string detailing the plan to validate the hypotheses.
- 'Data Used': A list of numbers denoting which data are utilized in the validation process, keep the number of them within 3. As textual data is hard to handle, please only select necessary textual data, and keep the number of them within 1.
- If the hypotheses are infeasible to validate, only include 'Feasibility' in the JSON output.

Table 17: Example prompt for feasibility check.

**Prompt for Hypothesis Validation**

Please write code to validate the following hypotheses using the provided data.
Hypotheses:
{hypotheses within the idea}
Data:
{metadata of datasets selected}

The last line of your output should be the final answer, in the JSON format like {'Hypothesis 1': 'Supported', ...}. The value for each hypothesis should be 'Supported' or 'Not supported'. If the evidence for the hypothesis is insignificant/mixed/limited/partial, the hypothesis is also classified as not supported.

Table 18: Example prompt for hypothesis validation.

---

**Prompt for Validation Process Summarization**

Here is the validation process of several hypotheses. It contains steps in both text and code formats. For steps in text format, the step contains keys 'type' and 'content'. For steps in code format, the step contains keys 'type', 'content', and 'output' or 'error'.

Please summarize the validation process in natural language, removing unnecessary steps and errors. Only keep the crucial reasoning process and results that lead to the final conclusion.

Your output should be a list in json structure. Each item in the list is a dict with keys 'type' and 'summarization'. The value of 'type' is 'text' or 'code', and the value of 'summarization' is a string describing the step. Limit the output into 1000 tokens.

Original Validation Steps:
{raw validation traces}

Output:

---

Table 19: Example prompt for validation process summarization.

---

**Prompt for Idea Selection in §5**

You are an expert researcher in political science. You are given two research ideas related to the topic {research topic}. Your task is to identify which idea is better from the following four dimensions 'Significance', 'Novelty', 'Feasibility', and 'Expected Effectiveness'.

Each research idea comprises the following four parts.
Research Question: A specific question about a behavior, event, or phenomenon of interest that the researcher wishes to seek answers for in the research.
Theory: Reasonably speculate on the answer to the research question, including a statement about why the proposed answer is correct.
Hypotheses: Identify observable implications of the theory, i.e., things we would observe if the theory is correct, and make predictions about relationships between measurable indicators of the theory's concepts.
*Preliminary Validation: Summarization of the preliminary validation process of the hypotheses.*

Evaluation Criteria:
{detailed content of the evaluation criteria}
Note: Please make your decision based on the weighted assessment of sub-criteria to avoid subjective bias. Avoid any position biases and ensure that the order of the two ideas does not influence your decision. DO NOT allow the LENGTH of the ideas to influence your evaluation. Be as objective as possible.

Here are the two research ideas for you to assess:
Idea 1:
{content of idea 1, *containing the summarized validation process*}
Idea 2:
{content of idea 2, *containing the summarized validation process*}
Please provide an explanation supporting your assessment. At the last line of your response, format your assessment in JSON with the keys: 'Significance', 'Novelty', 'Feasibility', and 'Expected Effectiveness'. The value of each key is an integer ranging from 0 to 2. 0 means a tie, 1 means idea 1 is better, and 2 means idea 2 is better.

---

Table 20: Example prompt for idea selection in §5, which differs from Table 16 in adding the validation results.

