# OpenReview forum: "Augmenting Research Ideation with Data: An Empirical Investigation in Social Science"
_ICLR.cc/2026/Conference — Submitted to ICLR 2026_

### Official Review · Reviewer_GJ8t · 2025-11-01

**Soundness:** 3
**Presentation:** 3
**Contribution:** 2
**Rating:** 4
**Confidence:** 4

**Summary:**

This paper explores the problem of low feasibility and effectiveness in research ideas generated by Large Language Models (LLMs). The authors propose a data-augmented ideation framework to improve idea quality, which introduces (1) metadata-guided idea generation—providing dataset descriptions to LLMs to guide feasible idea generation, and (2) automatic preliminary validation—allowing LLMs to conduct empirical checks on hypotheses using available data.
Experiments are conducted in the domain of climate negotiations. The authors construct a dataset collection called CLIMATEDATABANK, perform automatic and human evaluations on feasibility, novelty, and effectiveness, and further run a user study with researchers to assess inspiration effects. Results show improvements  compared to non-augmented baselines.

**Strengths:**

1. The paper is clearly written, logically organized, and easy to follow.
2. Applying LLM-based ideation to the social science domain is an interesting and relatively unexplored area.
3. The inclusion of a human study is commendable, as it goes beyond evaluating the LLM’s direct outputs and provides preliminary empirical evidence of the framework’s practical utility.
4. The creation of the CLIMATEDATABANK is a useful resource for future research in this specific social science domain.

**Weaknesses:**

1. The proposed data-augmented ideation is a relatively straightforward extension of existing frameworks. Adding dataset descriptions as metadata and performing simple validation are natural incremental steps, not a fundamentally new approach or theoretical contribution.
2. The automatic validation process is largely descriptive and based on keyword counts or correlations, not rigorous statistical or causal analysis. It is unclear how reliable or generalizable these validations are, and they do not convincingly demonstrate an improvement in true hypothesis verification.
3. All experiments are conducted in one social science topic (climate negotiation). This makes it difficult to generalize claims about research ideation in general.
4. The paper reveals an unaddressed trade-off: incorporating metadata enhances feasibility but reduces novelty, suggesting that the framework systematically biases idea generation toward safer, data-driven concepts at the expense of creativity—a limitation that remains insufficiently analyzed or discussed.

**Questions:**

1. The paper's main intervention seems to push LLMs towards "safer" ideas that are directly verifiable with the provided data, resulting in a drop in novelty. How do you see this framework mitigating the risk of simply generating obvious or incremental ideas that are "low-hanging fruit" in the data, rather than genuinely novel research directions?
2. The evaluation is conducted exclusively within the domain of climate negotiations. How generalizable is the proposed framework to other scientific disciplines, where data structures, hypothesis formulation, and validation standards differ substantially? Have the authors considered cross-domain experiments to support broader applicability?

---

> ### Author Response · Authors · 2025-11-25
> **Response to Reviewer GJ8t (1/2)**
>
> Thank you for your constructive comments! Below we address your concerns in detail.
>
> 1. Contribution of the paper
> > The proposed data-augmented ideation is a relatively straightforward extension of existing frameworks. Adding dataset descriptions as metadata and performing simple validation are natural incremental steps, not a fundamentally new approach or theoretical contribution.
>
> LLM-assisted research ideation is an important emerging direction, but it remains in an early stage. Therefore, we believe it is crucial to conduct rigorous empirical analyses of how different design choices, such as incorporating metadata and performing preliminary validation, affect this process. While it may be intuitive that these components could help, their actual influence was not known prior to systematic experimentation. Our work provides empirical evidence and insights that can inform and guide future research on LLM-based ideation.
>
> In addition, we examine whether LLM-generated ideas, together with their associated datasets and validation processes, can inspire human researchers to develop their own ideas. This highlights the practical value of LLM ideation and illustrates another benefit of automatic validation beyond idea selection: it provides concrete signals about how an idea can be implemented, and enables researchers to refine the preliminary validation into a more rigorous validation.
>
> 2. Reliability of automatic validation using LLMs
> > The automatic validation process is largely descriptive and based on keyword counts or correlations, not rigorous statistical or causal analysis. It is unclear how reliable or generalizable these validations are, and they do not convincingly demonstrate an improvement in true hypothesis verification.
>
> In practice, most of the automatic validation processes we observe involve statistical or causal methods, not just keyword or correlation checks. Among the 15 hypotheses examined in the human evaluation of our pilot study (Section 5.2), 10 are validated using statistical or causal analyses. An example is shown below:
>
> ```
> # Hypothesis 2: The effect of group membership becomes stronger over time.
> # Fit logistic regression with interaction between same_group and year
> logit_model = smf.logit('cooperation ~ same_group * year', data=data).fit(disp=0)
>
> # Test the interaction term for significance
> interaction_p_value = logit_model.pvalues['same_group:year']
>
> # Evaluate Hypothesis 2
> hypothesis_2_result = "Supported" if interaction_p_value < 0.05 else "Not Supported"
> ```
>
> Although the validation experiments in the main paper use GPT-4o with the code interpreter assistant, the framework itself is not tied to a particular validation method. With the emergence of stronger coding models such as Codex and Claude Code, automatic validation can become more reliable.
>
> For example, we ask Codex to review and correct GPT-4o’s validation processes. Among the 15 hypotheses, Codex correctly identifies 4 issues that are also flagged by human annotators, and after incorporating these corrections, the overall accuracy increased from 73% to 87%. For example, in an experiment, GPT-4o relies solely on correlations to validate a hypothesis, and Codex detects this flaw and successfully corrects it.
>
> 3. Experiment domain
> > The evaluation is conducted exclusively within the domain of climate negotiations. How generalizable is the proposed framework to other scientific disciplines, where data structures, hypothesis formulation, and validation standards differ substantially? Have the authors considered cross-domain experiments to support broader applicability?
>
> We add experiments in mechanistic interpretability, a domain from the AI discipline, with datasets **automatically collected by LLMs**. Because this experiment is extensive and all reviewers are interested in it, we include the details in the general comments. We kindly invite you to take a look.
>
> In brief, both automatic and human evaluations show that our data-augmented LLM ideation framework also improves idea quality in mechanistic interpretability, particularly in feasibility and expected effectiveness.
>
> We did not conduct extensive cross-domain experiments due to limited resources and the difficulty of recruiting expert annotators across multiple fields. However, we believe this is a promising direction, and that the *unusual combination of data from different domains* has the potential to spark new scientific insights [1].
>
> [1] Yu, Yulin, and Daniel M. Romero. "Does the use of unusual combinations of datasets contribute to greater scientific impact?." Proceedings of the National Academy of Sciences 121.41 (2024): e2402802121.

---

> ### Author Response · Authors · 2025-11-25
> **Response to Reviewer GJ8t (2/2)**
>
> 4. Balance between empirical tractability and theoretical originality
> > The paper's main intervention seems to push LLMs towards "safer" ideas that are directly verifiable with the provided data, resulting in a drop in novelty. How do you see this framework mitigating the risk of simply generating obvious or incremental ideas that are "low-hanging fruit" in the data, rather than genuinely novel research directions?
>
> We agree that, just as human researchers navigate tradeoffs between theoretical ambition and empirical tractability, LLMs also face a similar tension during ideation. In the following experiment, we show that researchers can steer LLMs toward more novel or more empirically tractable ideas by adjusting the prompts.
>
> In addition to our original metadata-augmented prompt, we conduct experiments with two prompt variants designed to modulate the tradeoff between creativity and practicality:
> * Flexible: `Note that these data are just for your reference, and there is no restriction that your hypotheses must be validated by these data.`
> * Strict: `Ensure the hypotheses in your idea are feasible to validate with the given data. Before finalizing your research idea, critically consider whether the data you have access to is sufficient to test your hypotheses.`
>
> We run automatic evaluation using Claude-3.5-Sonnet to compare idea generation across four settings: (1) without metadata, (2) with metadata (original prompt), and the two new variants. ELO scores are shown below:
>
> | **Method**                |  **Significance** | **Novelty** | **Feasibility** | **Exp. Effectiveness** | **Average** |
> |---------------------------------|----------------------|-----------------------|------------------|----------------------|-----------------------------|
> | w/o Metadata     |  1018  |  **1030** |  841 |  892 | 934 |
> | w. Metadata (flexible) |  1014              | 1014               |  1006              |  1026              |  1023              |
> | w. Metadata (original) |  **1042**              | 981               |  1066              |  1031              |  **1048**              |
> | w. Metadata (strict) |   925             |  976              |  **1087**              |   **1051**             |  995              |
>
> The results indicate:
> * Under the strict condition, feasibility and expected effectiveness are highest, but novelty declines.
> * Under the flexible condition, novelty is closer to the level of “w/o metadata”, while feasibility and expected effectiveness remain substantially higher.
> * The effect of metadata on significance is mixed: metadata may inspire the LLM to pursue impactful directions it previously overlooked, but it may also constrain the space of potential ideas.
>
> It is also worth noticing that *novelty in scientific research must rest on empirical tractability*: the goal is not unconstrained novelty, but novel ideas that can realistically be pursued and validated. Prior LLM ideation methods have been criticized for generating ideas that are difficult or impossible to implement and whose usefulness is uncertain. Our framework provides a practical way to steer LLMs toward ideas that are both novel and feasible, and these ideas are rated of better overall quality by expert annotators.

---

### Official Review · Reviewer_cDfR · 2025-11-01

**Soundness:** 3
**Presentation:** 3
**Contribution:** 2
**Rating:** 4
**Confidence:** 4

**Summary:**

This work proposes to augment the conventional LLM idea generation pipeline with two additional steps: (1) grounding the idea generation on specific dataset metadata; and (2) automated feasibility check for idea selection.

The experiments are done on the social science domain. For the metadata conditioning, they first constructed a dataset called ClimateDataBank consisting of 22 datasets in CSV format, 8 reference papers, and manually curated research topics.

Automatic and human evaluations show that adding metadata generally improves the feasibility of the generated ideas (especially from Table 3) where the difference in human preference is statistically significant.

Next, the authors incorporate an automated validation step, where an LLM selects the appropriate dataset and runs code in a sandbox environment to validate the generated hypothesis.

Automatic execution aligns with the ground truth conclusions 70% of the time, and humans generally find the automatic execution trajectories useful (Table 4b). Furthermore, ideas selected by the automatic valiation process are prferred by human experts across all metrics (Table 6).

**Strengths:**

- The authors did quite extensive human expert evaluation for all experiments in the paper, which makes the conclusions more convincing.

- The empirical results are positive, and human evaluators find the generated ideas helpful and inspiring.

- The proposed ideas (metadata conditioning and automatic validation) are extremely simple and easy to implement.

**Weaknesses:**

- My biggest concern is that all experiments are done on the social science domain (10 climate negotiation-related research topics in Appendix A), and I'm not sure whether the conclusions could be generalizable across other domains? For example, how would this work for empirical AI research? What would the automatic valiation look like in that case? My biased view is that the automatic validation is only possible for research problems where the execution is quite simple and straightforward, and you are gonna see a lot more errors when you move to a more execution-heavy domain, for example, LLM post-training/pre-training research, where the code implementation is more involved, and the execution involves GPUs as well.

- Looks like the metadata construction is purely manual right now, how is this scalable?

- My overall judgement is that it's great to see the proposed pipeline works for the social science topics being tested, but I'm not quite convinced it's gonna work the same for other broader domains. I'm thus leaning borderline reject unless the authors can convince me during rebuttal that they made this work in another domain too.

**Questions:**

N/A

---

> ### Author Response · Authors · 2025-11-25
> **Response to Reviewer cDfR**
>
> Thank you for your helpful comments! Below we address your concerns in detail.
>
> 1. Experiment domain
> > My biggest concern is that all experiments are done on the social science domain (10 climate negotiation-related research topics in Appendix A), and I'm not sure whether the conclusions could be generalizable across other domains? For example, how would this work for empirical AI research? What would the automatic validation look like in that case?
>
> We add experiments in mechanistic interpretability, a domain from the AI discipline. Because this experiment is extensive and all reviewers are interested in it, we include the details in the general comments. We kindly invite you to take a look.
>
> Specifically, we conduct the automatic validation using Codex, because the code interpreter assistant used for social science validation is not suitable for language model experiments (its sandbox does not support packages such as transformers). We place the selected datasets in Codex’s working directory and instruct it to run preliminary validation in a Jupyter Notebook, and, at the end of the validation, summarize the process in natural language using Markdown. We equip Codex with the Scribe MCP server to enhance the reliability of its interactions with the Jupyter environment.
>
> Importantly, the framework is not tied to a specific validation procedure. As LLMs continue to improve in autonomous experimentation and when more computational resources are accessible, this framework can extend to broader areas, such as LLM post-training/pre-training research.
>
> 2. Data collection
> > Looks like the metadata construction is purely manual right now, how is this scalable?
>
> In the mechanistic interpretability experiment, data collection is performed automatically by LLMs, and the resulting datasets prove useful for both idea generation and idea selection. This demonstrates that in domains such as AI, where open-source datasets are widely available, the dataset collection step can also be automated, making the overall framework more scalable.

---

> > ### Comment · Reviewer_cDfR · 2025-11-25
> >
> > Thanks for the response. The additional experiment is cool, and it's impressive that you managed to put them together in a short period of time. I think this does resolve the generalizability concern in my original review.
> >
> > But on a more fundamental level, I think the idea of including metadata information of datasets and preliminary simulated experiments for idea selection is rather intuitive -- it's great to see empirical validation that this works, but honestly, it's not surprising at all that it's better than the baseline of not having anything.
> >
> > From a technical perspective, the practice of using a coding agent to run validation experiments is not a new idea at all. Previous works such as "Data-driven Discovery with Large Generative Models" and "The AI Scientist: Towards Fully Automated Open-Ended Scientific Discovery" have done different versions of this, including the entire ideation-execution pipeline.
> >
> > So despite the great efforts, I'm still leaning towards my original score given the technical contributions in the contexts of various prior works on automated ideation and fully automated research workflows.

---

> > > ### Author Response · Authors · 2025-11-26
> > >
> > > Thank you for the response! We're also glad to hear that the additional experiment addresses your original concerns.
> > >
> > > Regarding the new point about the technical contribution of our paper, we would like to offer a few clarifications:
> > >
> > > **1. Importance of the empirical study.**
> > >
> > > Although it may seem intuitive that incorporating metadata and running preliminary validations would help, their actual influence is not known prior to systematic experimentation. In an emerging research direction like LLM-assisted research ideation, we believe that careful empirical analysis of such design choices is essential. Our work provides concrete evidence and insights into how and to what extent these components matter, which can inform and guide future research on LLM-based ideation.
> > >
> > > **2. Positioning relative to prior work on automated experiments.**
> > >
> > > While our approach involves code generation and execution, its purpose is not to serve as a full experimental pipeline. Instead, these executions function as lightweight validation signals to support idea selection. This is distinct from prior systems such as *The AI Scientist*, which aim for full ideation–execution autonomy. On the other hand, *Data-driven Discovery with Large Generative Models* is a position paper proposing a conceptual blueprint. Our contribution is more concrete: we present a framework for incorporating data into the ideation process and empirically demonstrate how data-driven validation influences idea quality.
> > >
> > > **3. Influence on human researchers.**
> > >
> > > A novel aspect of our work is studying whether LLM-generated ideas, paired with their associated datasets and preliminary validations, can inspire human researchers to develop their own ideas. This highlights an important practical dimension of LLM-assisted ideation. Beyond filtering ideas, automatic validation offers actionable signals about how an idea could be implemented, enabling researchers to refine preliminary validation into more rigorous testing. This demonstrates the value of data-driven ideation in supporting human–AI collaborative research.
> > >
> > > We hope these clarifications help contextualize our contributions within the broader landscape of automated ideation and research workflows.

---

### Official Review · Reviewer_Dwgf · 2025-11-04

**Soundness:** 4
**Presentation:** 4
**Contribution:** 4
**Rating:** 6
**Confidence:** 4

**Summary:**

This paper proposes a data-augmented framework to enhance Large Language Model (LLM) research ideation, specifically addressing the lack of feasibility and effectiveness in purely literature-driven ideas. The framework introduces dataset metadata during the idea generation stage to guide feasibility, and an automated preliminary validation step during idea selection to confirm empirical plausibility. Experiments in the social science domain (climate negotiations) show that metadata significantly improves feasibility (by 20%) and expected effectiveness, while automated validation enhances the overall quality of selected ideas. Furthermore, a human study demonstrates that these augmented LLM-generated ideas successfully inspire human researchers to propose higher-quality research.

**Strengths:**

1. The paper provides a novel, two-pronged framework that effectively integrates empirical data signals (metadata and preliminary validation) directly into the LLM ideation pipeline, which is a significant step beyond existing literature-based approaches.
2. The results are robust, supported by both automatic evaluations using multiple LLM judges and a controlled human expert evaluation that confirms substantial gains in feasibility (20%) and expected effectiveness (18%).
3. The human study successfully validates the utility of the LLM-generated ideas, showing that they are not just quantitatively superior but also serve as a useful source of inspiration, leading researchers to propose superior ideas in practice.

**Weaknesses:**

1. The investigation is strictly confined to one niche domain (quantitative social science on climate negotiations) using a custom-built dataset (CLIMATEDATABANK). The framework’s transferability to other scientific disciplines or fields with less structured data remains unproven.
2. The data-aware generation process appears to impose an implicit constraint on creativity, resulting in a reported decline in the novelty metric for some experimental settings. This suggests a need to better manage the balance between empirical tractability and theoretical originality.
3. The automated validation step depends on the LLM’s ability to generate and execute correct code for hypothesis testing. The robustness and reliability of this LLM-as-statistician module are not deeply quantified (e.g., code error rate, statistical inference quality), posing a potential point of fragility.

**Questions:**

1. How would the data-augmented framework be adapted for and evaluated in research domains that primarily rely on qualitative data (e.g., interview transcripts, ethnographic notes) rather than the structured textual/panel/cross-sectional data currently in the CLIMATEDATABANK?
2. To mitigate the noted drop in novelty, could the authors introduce a tunable parameter in the prompt (e.g., a "creativity vs. practicality" score) to explicitly control the LLM's adherence to the provided metadata, allowing researchers to explore a wider spectrum of ideas?
3. Given the innovative but complex nature of LLM-driven automatic validation, what is the observed error rate (e.g., code execution failure, incorrect statistical conclusion) of the validation step, and what safeguards (e.g., code environment constraints, second-pass LLM review) are in place to ensure the validation signals are dependable?

---

> ### Author Response · Authors · 2025-11-25
> **Response to Reviewer Dwgf (1/2)**
>
> Thank you for your constructive comments! Below we address your concerns in detail.
>
> 1. Experiment domain
> > The investigation is strictly confined to one niche domain (quantitative social science on climate negotiations) using a custom-built dataset (CLIMATEDATABANK). The framework’s transferability to other scientific disciplines or fields with less structured data remains unproven.
>
> We add experiments in mechanistic interpretability, a domain from the AI discipline, with datasets **automatically collected by LLMs**. Because this experiment is extensive and all reviewers are interested in it, we include the details in the general comments. We kindly invite you to take a look.
>
> In brief, both automatic and human evaluations show that our data-augmented LLM ideation framework also improves idea quality in mechanistic interpretability, particularly in feasibility and expected effectiveness.
>
> We did not experiment with qualitative domains because relevant data, such as interview transcripts, are typically not publicly available and are harder to collect. However, we believe that if LLMs are granted access to such data, they could also generate meaningful research ideas and perform corresponding analyses. In these settings, the validation process may shift away from code-based experiments toward more interpretive or analytical forms of deep research.
>
> 2. Balance between empirical tractability and theoretical originality
> > To mitigate the noted drop in novelty, could the authors introduce a tunable parameter in the prompt (e.g., a "creativity vs. practicality" score) to explicitly control the LLM's adherence to the provided metadata, allowing researchers to explore a wider spectrum of ideas?
>
> In addition to our original metadata-augmented prompt, we conduct experiments with two prompt variants designed to modulate the tradeoff between creativity and practicality:
> * Flexible: `Note that these data are just for your reference, and there is no restriction that your hypotheses must be validated by these data.`
> * Strict: `Ensure the hypotheses in your idea are feasible to validate with the given data. Before finalizing your research idea, critically consider whether the data you have access to is sufficient to test your hypotheses.`
>
> We run automatic evaluation using Claude-3.5-Sonnet to compare idea generation across four settings: (1) without metadata, (2) with metadata (original prompt), and the two new variants. ELO scores are shown below:
>
> | **Method**                |  **Significance** | **Novelty** | **Feasibility** | **Exp. Effectiveness** | **Average** |
> |---------------------------------|----------------------|-----------------------|------------------|----------------------|-----------------------------|
> | w/o Metadata     |  1018  |  **1030** |  841 |  892 | 934 |
> | w. Metadata (flexible) |  1014              | 1014               |  1006              |  1026              |  1023              |
> | w. Metadata (original) |  **1042**              | 981               |  1066              |  1031              |  **1048**              |
> | w. Metadata (strict) |   925             |  976              |  **1087**              |   **1051**             |  995              |
>
> The results indicate:
> * Under the strict condition, feasibility and expected effectiveness are highest, but novelty declines.
> * Under the flexible condition, novelty is closer to the level of “w/o metadata”, while feasibility and expected effectiveness remain substantially higher.
> * The effect of metadata on significance is mixed: metadata may inspire the LLM to pursue impactful directions it previously overlooked, but it may also constrain the space of potential ideas.
>
> These findings mirror how human researchers navigate tradeoffs between theoretical ambition and empirical tractability when developing research ideas. LLM ideation exhibits similar dynamics. In practice, researchers can adjust prompts according to their preferred balance. It would also be interesting to explore whether model-steering techniques can more accurately position LLMs along the creativity–tractability curve, or finding an optimal point for each research topic.

---

> ### Author Response · Authors · 2025-11-25
> **Response to Reviewer Dwgf (2/2)**
>
> 3. Reliability of automatic validation using LLMs
> > Given the innovative but complex nature of LLM-driven automatic validation, what is the observed error rate (e.g., code execution failure, incorrect statistical conclusion) of the validation step, and what safeguards (e.g., code environment constraints, second-pass LLM review) are in place to ensure the validation signals are dependable?
>
> We conduct a pilot study of hypothesis-validation performance in Section 5.2. The results show that GPT-4o with the code interpreter assistant achieves over 70% accuracy when validating hypotheses drawn from existing papers. Human evaluation further indicates that half of the validation processes largely support the hypotheses with only minor flaws, while another 40% partially align with the hypotheses but raise substantial concerns, for example, insufficient control variables. Data analysis, especially involving textual data, is the most challenging component. Other issues, including knowledge recall, reasoning-code generation, and result interpretation, also appear in roughly 30% of cases.
>
> Despite the imperfections, annotators note that the automatic validation is *helpful as an auxiliary tool for exploratory research*, which aligns well with our intended use of the validation process: as a reference signal for idea selection rather than a rigorous statistical test.
>
> The validation takes place in the internal sandbox of the code interpreter assistant, ensuring isolation from the external environment and preventing any unintended effects on the user’s code environment.
>
> To evaluate whether a second-pass LLM review can further improve the validation reliability, we conduct an additional experiment in which Codex reviews each validation process and attempts to correct any identified flaws. Across 15 hypotheses from 6 papers, Codex correctly flags 4 issues that are also noted by human annotators (e.g., insufficient covariate coverage, and using only correlations to validate causal hypotheses). After incorporating these corrections, the accuracy increases from 73% to 87%, suggesting that reviewing with a strong LLM is a promising way to enhance the reliability of the validation.

---

### Author Response · Authors · 2025-11-25
**General Response: Experiments in the Mechanistic Interpretability Domain (1/4)**

We conduct experiments in another research domain: mechanistic interpretability, to demonstrate the generalizability of our method.

We choose this domain for two reasons: (1) It is substantially different from the climate-negotiation domain used in our main experiments, making it a strong testbed for assessing whether our framework can transfer across scientific disciplines; and
(2) a considerable number of  ideas in mechanistic interpretability can be experimented on small language models, which makes it feasible to conduct preliminary automatic validation with limited resources.

The experiments span five subtopics within mechanistic interpretability:
| **Topic** | **Description** |
|--|--|
| Circuit Discovery for Mechanistic Interpretability | Identifying the specific attention and MLP components that jointly implement a model’s internal computation. |
| Causal Tracing for Mechanistic Interpretability | Mapping how information flows through a model by tracking and intervening on intermediate representations. |
| Mechanistic Interpretability of Knowledge Learning | Understanding how language models store and retrieve factual information through identifiable circuits or features. |
| Mechanistic Interpretability of Arithmetic Reasoning | Understanding the internal algorithms models use to perform numerical or multi-step arithmetic tasks. |
| Mechanistic Interpretability for Safety | Applying mechanistic insights to detect, understand, and mitigate harmful or unintended internal computations. |

## 1. Data collection
Unlike the manual construction of ClimateDataBank, here we rely on LLMs to automate data collection. Specifically, we instruct Codex to search the web and collect 20 datasets that could support mechanistic interpretability research, provide a one-sentence description for each, and download them to a designated folder.

The resulting dataset list is included in the updated supplementary material. It contains mechanistic-interpretability-specific datasets such as `fahamu/ioi` and `mib-bench/arithmetic_addition`, as well as more general datasets used to evaluate language-model capabilities, such as `allenai/ZebraLogicBench` and `scan`. Although some chosen datasets are similar, the full list covers a broad spectrum of data, and we use it directly for subsequent experiments.

---

## 2. Incorporating metadata in idea generation
We use GPT-Researcher as our baseline method, as it performs best in the main experiment of our paper. The underlying LLM used is GPT-4o. For each of the five topics, we instruct GPT-Researcher to generate 20 ideas with metadata and 20 ideas without metadata. It then selects the top five ideas in each setting using a Swiss tournament.


For automatic evaluation, we employ Gemini-2.5-Flash (replacing Gemini-1.5-Pro in the main experiment, which is now deprecated) and Claude-3.5-Sonnet as judge models. For human evaluation, two AI researchers with PhDs annotate the 25 paired ideas.

Automatic evaluation results (ELO scores):
| **Judge**                | **w. Metadata** | **Significance** | **Novelty** | **Feasibility** | **Exp. Effectiveness** | **Average** |
|---------------------------------|----------------------|-----------------------|------------------|----------------------|-----------------------------|------------------|
| Gemini-2.5-Flash  | ✗              |  995                  |  957    | 993               |  986                        | 954              |
|                                 | ✓               | **1005**          |  **1043**             |  **1007**       |  **1014**           | **1046**     |
| Claude-3.5-Sonnet  | ✗              |  979                  |  986    | 990               |  993                        |  979             |
|                                 | ✓               |  **1021**         | **1014**              | **1010**        |  **1007**           |  **1021**    |

Human evaluation results:
|                | **w. Metadata** | **Tie** | **w/o Metadata** |
|--------------------|--------------------------|------------------|---------------------------|
| Significance       |  **40**           | 24             |  36           |
| Novelty            |  **36**           | 36           | 28            |
| Feasibility        |  **44**           | 40             | 16            |
| Exp. Effectiveness |  **44**           | 36             | 20           |
| Overall            |  **52**           | 20             | 28            |

Both automatic and human evaluations consistently show that incorporating metadata improves ideation quality across all dimensions. In human evaluation, the improvements are substantial in feasibility and expected effectiveness, with additional gains in significance and novelty.

---

> ### Author Response · Authors · 2025-11-25
> **General Response: Experiments in the Mechanistic Interpretability Domain (2/4)**
>
> Below we demonstrate representative ideas generated for the topic `Mechanistic Interpretability of Arithmetic Reasoning`, with and without metadata.
>
> With metadata:
> ```
> Research Question: Do transformer-based language models implicitly learn algebraic structures (e.g., commutativity, distributivity) during arithmetic reasoning, and how are these structures represented internally?
> Theory: Transformer models may develop internal representations that encode algebraic properties, such as commutativity (e.g., a + b = b + a) and distributivity (e.g., a × (b + c) = a × b + a × c), as a result of training on arithmetic datasets. These representations could emerge in attention patterns or MLP activations, enabling the models to generalize to unseen arithmetic tasks.
> Hypotheses:
> Algebraic properties, such as commutativity, will be represented as invariant attention patterns across different permutations of input operands.
> Distributive relationships will manifest as distinct activation pathways in MLP layers, corresponding to the decomposition of arithmetic expressions.
> Probing experiments will reveal that models trained on arithmetic datasets generalize better to unseen tasks involving algebraic reasoning compared to models trained on generic datasets.
> Intervening on specific model components (e.g., attention heads) will disrupt algebraic reasoning, confirming their role in encoding algebraic structures.
> ```
>
> Without metadata:
> ```
> Research Question: How do different layers of transformer-based LLMs specialize for specific arithmetic subtasks, such as addition, subtraction, multiplication, and division?
> Theory: Different layers in a transformer model specialize for distinct arithmetic subtasks, with early layers focusing on embedding operands and operators, middle layers performing intermediate computations, and later layers synthesizing the final results. This hierarchical specialization enables the model to handle complex, multi-step arithmetic tasks.
> Hypotheses:
> Early layers primarily encode operand and operator embeddings, while middle layers perform the bulk of arithmetic computations.
> Models fine-tuned on specific arithmetic tasks will exhibit stronger layer specialization compared to general-purpose models.
> Visualization of attention weights and neuron activations will reveal distinct patterns for addition, subtraction, multiplication, and division.
> ```
> Compared to the idea generated with metadata, the idea generated without metadata is more general and lacks deeper insights. Here, metadata effectively guides the LLM to engage with the topic at a more substantive conceptual level.
>
> ---
>
> ## 3. Incorporating automatic validation in idea selection
> For the ideas generated by GPT-Researcher with metadata, we first assess the feasibility of validating their hypotheses using GPT-4o, following a procedure similar to that described in Section 5.1. If GPT-4o determines that the hypotheses are testable, it outputs the indices of up to three relevant datasets along with a corresponding validation plan.
>
> For all feasible ideas, we then conduct preliminary hypothesis validation using Codex. (In the main paper, we use GPT-4o equipped with the Code Interpreter assistant, which is well suited for data analysis but unsuitable for language model experiments, as the execution sandbox does not support packages such as transformers.) We place the selected datasets in Codex’s working directory and instruct it to run preliminary validation in a Jupyter Notebook, and, at the end of the validation, summarize the process in natural language using Markdown. We equip Codex with the Scribe MCP server (https://github.com/goodfire-ai/scribe/) to enhance the reliability of its interactions with the Jupyter environment. To ensure efficiency, we encourage Codex to perform experiments using GPT-2 and limit the total validation time within 20 minutes.
>
> After validation is completed, we ask GPT-4o to select the top five ideas for each topic with access to the validation summaries. We then conduct automatic and human evaluations to compare ideas selected with vs. without validation, following the same evaluation setup used in the metadata experiment.

---

> ### Author Response · Authors · 2025-11-25
> **General Response: Experiments in the Mechanistic Interpretability Domain (3/4)**
>
> Automatic evaluation results (ELO scores):
> | **Judge**                | **w. Validation** | **Significance** | **Novelty** | **Feasibility** | **Exp. Effectiveness** | **Average** |
> |---------------------------------|----------------------|-----------------------|------------------|----------------------|-----------------------------|------------------|
> | Gemini-2.5-Flash  | ✗              |  **1008**                  |  **1000**   | 981               |  977                        | 992             |
> |                                 | ✓               | 992          |  **1000**             |  **1019**       |  **1023**           | **1008**     |
> | Claude-3.5-Sonnet  | ✗              |  **1003**                  |  986    | 986               |  976                        |  993             |
> |                                 | ✓               |  997         | **1014**              | **1014**        |  **1024**           |  **1007**    |
>
> Human evaluation results:
> |                | **w. Validation** | **Tie** | **w/o Validation** |
> |--------------------|--------------------------|------------------|---------------------------|
> | Significance       |  28           |  40            |  **32**           |
> | Novelty            |  **36**           |  32            |  32           |
> | Feasibility        |  **32**           |  40            |  28           |
> | Exp. Effectiveness |  **36**           |  44            |  20           |
> | Overall            |  **40**           |  28            |  32           |
>
> Both automatic and human evaluations show similar trends: incorporating validation substantially improves the expected effectiveness of selected ideas, with only minor reductions in significance. Overall, validated ideas are judged to be stronger. Although these experiments are preliminary and rely on small language models, the validation process helps the LLM engage more deeply with each idea and provides meaningful evidence of its promise.
>
> Below we show the validation summary for the earlier example idea generated with metadata. For readability, only the natural-language summary is included here; the full validation code is available in the updated supplementary material.
> ```
> ## Experimental Setup
> - **Model:** GPT-2-small (pretrained) with additional fine-tuned copies (arithmetic vs WikiText) for probing.
> - **Datasets:** mib-bench/arithmetic_addition (train/val), mib-bench/arithmetic_subtraction (val), WikiText-2-raw (train split).
> - **Methods:** (H1) Commutativity assessed via operand-conditioned attention-mass cosine similarity across operand permutations. (H2) Registered hooks on MLP blocks to compare activation norms for combined vs. distributed expressions. (H3) Fine-tuned GPT-2-small on arithmetic vs. WikiText corpora and measured masked label losses on held-out addition/subtraction prompts. (H4) Zeroed individual attention heads (and a critical pair) identified in H1 while recomputing label losses.
>
> ## Key Findings
>
> ### H1 – Commutativity as Invariant Attention (Supported)
> - Average cosine similarity of operand-target attention distributions stays high across permutations (layer means ≥0.69 with layer-0 heads {1,3,5} ≈0.99).
> - Indicates the same heads preserve operand-specific routing regardless of operand order, aligning with invariant attention circuitry.
>
> ### H2 – Distributive Structure in MLP Pathways (Partially Supported)
> - Comparing `a * (b + c)` vs. `a*b + a*c` reveals large relative activation differences concentrated in mid/late MLPs (layers 5–9 show 0.51–0.68 relative deltas, layer 9 abs diff ≈21.7).
> - Confirms distinct activation pathways emerge when expressions are decomposed, though early layers (<4) show smaller separation (<0.38), so evidence is strongest for deeper blocks.
>
> ### H3 – Probes Favor Arithmetic-Tuned Models (Supported)
> - Masked label losses (lower is better) on 150 addition prompts: arithmetic-tuned = 8.01 vs. pretrained = 15.20 vs. WikiText-tuned = 17.52.
> - On 150 subtraction prompts (unseen task), arithmetic-tuned still leads (9.86) while WikiText-tuned degrades to 15.44, demonstrating better algebraic generalization for probes trained with arithmetic supervision.
>
> ### H4 – Head Interventions Disrupt Reasoning (Supported)
> - Zeroing head (layer 0, head 11) — one of the invariant commutative heads — increases addition loss by +0.135 (≈0.88%).
> - Removing heads (0,10)+(0,11) jointly raises loss by +0.173 (≈1.13%), whereas random heads show negligible or even negative impact, tying these specific heads to algebraic structure encoding.
>
> ## Overall Conclusion
> Targeted analyses show GPT-2 already encodes arithmetic regularities: attention heads in early layers act invariantly under operand swaps, mid/late MLPs differentiate distributed forms, and arithmetic-focused training sharply improves algebraic probes relative to generic LM tuning. Disabling the identified heads measurably harms arithmetic predictions, reinforcing their causal role in representing algebraic structure.
> ```

---

> ### Author Response · Authors · 2025-11-25
> **General Response: Experiments in the Mechanistic Interpretability Domain (4/4)**
>
> ## 4. Summary
> This experiment demonstrates that the data-augmented LLM ideation framework is effective not only for quantitative social science research but also for AI research domains where validation is inherently more complex. Moreover, in domains such as AI where rich open-source datasets are readily available, dataset collection itself can be automated by LLMs.
>
> Importantly, the framework is not tied to a specific validation procedure. As LLMs continue to improve in autonomous experimentation and when more computational resources are accessible, this framework can extend to broader scientific domains.
>
> We will include this experiment in the updated version of the paper.

---

### Author Response · Authors · 2025-12-03
**Summary of Rebuttal**

We thank all reviewers for their thoughtful and constructive reviews!

Strengths:
- Reviewer `Dwgf` highlights the *novelty* of the paper, emphasizing that our data-augmented ideation framework marks a significant step over purely literature-driven approaches.
- Reviewers `Dwgf` and `cDfR` highlight the *empirical results*, noting that the automatic and human evaluations provide convincing evidence of substantial improvements in idea quality.
- All reviewers praise that the *human study* demonstrates the practical utility and inspirational value of the LLM-generated ideas.
- Reviewer `GJ8t` underscores the contribution of exploring an unaddressed domain (quantitative social science) for ideation research and creating new resources that support future research in this area.

We carefully address all reviewer comments in the rebuttal, and summarize key additions and clarifications below:
- We added experiments in a new domain (mechanistic interpretability) to address generalizability concerns, showing that data-augmented ideation improves idea quality in this domain. This experiment also demonstrated that the dataset collection step can be automated by LLMs when rich open-source data is available.
- We experimented with strict and flexible metadata-conditioning prompt variants to study the creativity–tractability tradeoff.
- We clarified the contribution of our paper in novelly incorporating data into the *ideation process*, providing rigorous empirical analysis, and studying the inspiration influence of LLM-generated ideas on human researchers.

---

### Meta-Review · Area_Chair_7BqE · 2026-01-12

**Summary:**

This paper proposes a data-augmented framework for LLM-based research ideation, aimed at improving the feasibility and effectiveness of generated ideas by integrating dataset metadata during idea generation and performing automated preliminary validation during selection. Evaluated in the social science domain of climate negotiations using a newly curated resource—ClimateDataBank (22 datasets, 8 papers, and annotated topics)—the approach demonstrates that metadata conditioning boosts idea feasibility by ~20% (with statistically significant human preference), while automated validation (via sandboxed code execution) aligns with ground-truth conclusions 70% of the time and yields ideas consistently preferred by human experts. A user study further shows that these augmented ideas successfully inspire researchers to develop higher-quality proposals. The work offers a practical, empirically grounded pipeline that moves beyond purely literature-driven ideation toward actionable, data-informed scientific discovery. However, the evaluation of the paper should be significantly improved.

**Reviewer Scores:**

NA

---

### Decision · Program_Chairs · 2026-01-26

Reject